# Understanding Bandits with Graph Feedback

**Houshuang Chen**
Shanghai Jiao Tong University
chenhoushuang@sjtu.edu.cn

**Zengfeng Huang**
Fudan University
huangzf@fudan.edu.cn

**Shuai Li**
Shanghai Jiao Tong University
shuaili8@sjtu.edu.cn

**Chihao Zhang**
Shanghai Jiao Tong University
chihao@sjtu.edu.cn

## Abstract

The bandit problem with graph feedback, proposed in [Mannor and Shamir, NeurIPS 2011], is modeled by a directed graph $G = (V, E)$ where $V$ is the collection of bandit arms, and once an arm is triggered, all its incident arms are observed. A fundamental question is how the structure of the graph affects the min-max regret. We propose the notions of the fractional weak domination number $\delta^*$ and the $k$-packing independence number capturing upper bound and lower bound for the regret respectively. We show that the two notions are inherently connected via aligning them with the linear program of the weakly dominating set and its dual — the fractional vertex packing set respectively. Based on this connection, we utilize the strong duality theorem to prove a general regret upper bound $O\left(\left(\delta^* \log |V|\right)^{\frac{1}{3}} T^{\frac{2}{3}}\right)$ and a lower bound $\Omega\left(\left(\delta^*/\alpha\right)^{\frac{1}{3}} T^{\frac{2}{3}}\right)$ where $\alpha$ is the integrality gap of the dual linear program. Therefore, our bounds are tight up to a $(\log |V|)^{\frac{1}{3}}$ factor on graphs with bounded integrality gap for the vertex packing problem including trees and graphs with bounded degree. Moreover, we show that for several special families of graphs, we can get rid of the $(\log |V|)^{\frac{1}{3}}$ factor and establish optimal regret.

## 1 Introduction

The multi-armed bandit is an extensively studied problem in reinforcement learning. Imagining a player facing an $n$-armed bandit, each time the player pulls one of the $n$ arms and incurs a loss. At the end of each round, the player receives some feedback and tries to make a better choice in the next round. The expected regret is defined by the difference between the player's cumulative losses and cumulative losses of the single best arm during $T$ rounds. In this article, we assume the loss at each round is given in an adversarial fashion. This is called the *adversarial bandit* in the literature. The difficulty of the adversarial bandit problem is usually measured by the min-max regret which is the expected regret of the best strategy against the worst possible loss sequence.

Player's strategy depends on how the feedback is given at each round. One simple type of feedback is called *full feedback* where the player can observe all arm's losses after playing an arm. An important problem studied in this model is *online learning with experts* [14, 17]. Another extreme, introduced in [8], is the vanilla *bandit feedback* where the player can only observe the loss of the arm he/she just pulled. Optimal bounds for the regret, either in $n$ or in $T$, are known for both types of feedback.

The work of [24] initialized the study on the generalization of the above two extremes, that is, the feedback consists of the losses of a collection of arms. This type of feedback can be naturally described by a *feedback graph $G$* where the vertex set is $[n]$ and a directed edge $(i, j)$ means pulling

the arm $i$ can observe the loss of arm $j$. Therefore, the "full feedback" means that $G$ is a clique with self-loops, and the "vanilla bandit feedback" means that $G$ consists of $n$ disjoint self-loops.

A natural yet challenging question is how the graph structure affects the min-max regret. The work of [1] systematically investigated the question and proved tight regret bounds in terms of the time horizon $T$. They show that, if the graph is "strongly observable", the regret is $\Theta(T^{\frac{1}{2}})$; if the graph is "weakly observable", the regret is $\Theta(T^{\frac{2}{3}})$; and if the graph is "non-observable", the regret is $\Theta(T)$. Here the notions of "strongly observable", "weakly observable" and "non-observable" roughly indicate the connectivity of the feedback graph and will be formally defined in Section 2. However, unlike the case of "full feedback" or "vanilla bandit feedback", the dependency of the regret on $n$, or more generally on the structure of the graph, is still not well understood. For example, for "weakly observable" graphs, an upper bound and a lower bound of the regret in terms of the weak domination number $\delta(G)$ were proved in [1], but a large gap exists between the two. This suggests that the weak domination number might not be the correct parameter to characterize the regret.

We make progress on this problem for "weakly observable" graphs. This family of graphs is general enough to encode almost all feedback patterns of bandits. We introduce the notions of the fractional weak domination number $\delta^*(G)$, the $k$-packing independence number and provide evidence to show that they are the correct graph parameters. The two parameters are closely related and help us to improve the upper bound and lower bound respectively. As the name indicated, $\delta^*(G)$ is the fractional version of $\delta(G)$, namely the optimum of the linear relaxation of the integer program for the weakly dominating set. We observe that this graph parameter has already been used in an algorithm for "strongly observable" graphs in [3], where it functioned differently. In the following, when the algorithm is clear from the context, we use $R(G, T)$ to denote the regret of the algorithm on the instance $G$ in $T$ rounds. Our main algorithmic result is:

**Theorem 1.** *There exists an algorithm such that for any weakly observable graph, any time horizon* $T \geq n^3 \log(n)/\delta^{*2}(G)$, *its regret satisfies* $R(G, T) = O\left( (\delta^*(G) \log n)^{\frac{1}{3}} T^{\frac{2}{3}} \right)$.

Note that the regret of the algorithm in [1] satisfies $R(G, T) = O\left( (\delta(G) \log n)^{\frac{1}{3}} T^{\frac{2}{3}} \right)$. The fractional weak domination number $\delta^*$ is always no larger than $\delta$, and the gaps between the two can be as large as $\Theta(\log n)$. We will give an explicit example in Section 4.3 in which the gap matters and our algorithm is optimal. Theorem 1 can be seamlessly extended to more general time-varying graphs and probabilistic graphs. The formal definitions of these models are in Appendix E.

On the other hand, we investigate graph structures that can be used to fool algorithms. We say a set $S$ of vertices is a $k$-packing independent set if $S$ is an independent set and any vertex has at most $k$ out-neighbors in $S$. We prove the following lower bound:

**Theorem 2.** *Let* $G = (V, E)$ *be a directed graph. If* $G$ *contains a* $k$-packing independent set $S$ with $|S| \geq 2$, *then for any randomized algorithm and any time horizon* $T$, *there exists a sequence of loss functions such that the expected regret is* $\Omega\left( \max\left\{ \frac{|S|}{k}, \log |S| \right\}^{\frac{1}{3}} \cdot T^{\frac{2}{3}} \right)$.

For every $k \in \mathbb{N}$, we use $\zeta_k$, the $k$-packing independence number, to denote the size of the maximum $k$-packing independent set. To prove Theorem 2, we reduce the problem of minimizing regret to *statistical hypothesis testing* for which powerful tools from information theory can help.

We can use Theorem 2 to strengthen lower bounds in [1]. Besides, we show that large $\delta^*$ usually implies large $\zeta_1$ via studying the linear programming dual of fractional weakly dominating sets and applying a novel rounding procedure. This is also one of our main technical contributions. Combinatorially, the dual linear program is to find the maximum fractional vertex packing set in the graph. Specifically, we can establish lower bounds in terms of $\delta^*$ by applying Theorem 2:

**Theorem 3.** *If* $G$ *is weakly observable, then for any algorithm and any sufficiently large time horizon* $T \in \mathbb{N}$, *there exists a sequence of loss functions such that* $R(G, T) = \Omega\left( \left( \frac{\delta^*}{\alpha} \right)^{1/3} \cdot T^{\frac{2}{3}} \right)$, *where* $\alpha$ *is the integrality gap of the linear program for vertex packing.*

Clearly the exact lower bound is determined by the integrality gap of a certain linear program. In general graphs, we have a universal upper bound $\alpha = O\left( n/\delta^* \right)$. For concrete instances, we can

obtain clearer and tighter bounds on $\alpha$. For example, the linear program has a constant integrality gap $\alpha$ on graphs of bounded degree.

**Corollary 4.** *Let $\Delta \in \mathbb{N}$ be a constant and $\mathcal{G}_\Delta$ be the family of graphs with maximum in-degree $\Delta$. Then for every weakly observable $G = (V, E) \in \mathcal{G}_\Delta$, any algorithm and any sufficiently large time horizon $T \in \mathbb{N}$, there exists a sequence of loss functions such that $R(G,T) = \Omega((\delta^*)^{\frac{1}{3}} \cdot T^{\frac{2}{3}})$.*

We also show that for 1-degenerate directed graphs (formally defined in Section 2.1), the integrality gap is 1. This family of graphs includes trees and directed cycles. As a consequence, we have

**Corollary 5.** *Let $G$ be a 1-degenerate weakly observable graph. Then for any algorithm and any sufficiently large time horizon $T \in \mathbb{N}$, there exists a sequence of loss functions such that $R(G,T) = \Omega((\delta^*)^{\frac{1}{3}} \cdot T^{\frac{2}{3}})$.*

**Comparison of previous results and our results**

In Table 1, we compare our new upper bounds, lower bounds and their gap with previous best results.

Table 1: A comparison of results

| Graph Type | Previous best results [1] | | This work | |
|---|---|---|---|---|
| | Min-max regret | Gap | Min-max regret | Gap |
| General graphs | $O\left((\delta \log n)^{\frac{1}{3}} \cdot T^{\frac{2}{3}}\right)$ $\Omega\left(\max\left\{(\frac{\delta}{\log^2 n})^{\frac{1}{3}}, 1\right\} \cdot T^{\frac{2}{3}}\right)$ | See discussion below | $O\left((\delta^* \log n)^{\frac{1}{3}} \cdot T^{\frac{2}{3}}\right)$ $\Omega\left(\max\left\{(\frac{\delta^*}{\alpha})^{\frac{1}{3}}, 1\right\} \cdot T^{\frac{2}{3}}\right)$ | See discussion below |
| | for $\delta = \log^2 n$: $\Omega\left(T^{\frac{2}{3}}\right)$ | $O(\log n)$ | for $\delta = \log^2 n$: $\Omega\left(\log\log n \cdot T^{\frac{2}{3}}\right)$ | $O\left(\frac{\log n}{\log\log n}\right)$ |
| Trees / Bounded in-degree | Same as general graphs | $O(\log n)$ | $O\left((\delta^* \log n)^{1/3} \cdot T^{2/3}\right)$ $\Omega\left((\delta^*)^{1/3} \cdot T^{2/3}\right)$ | $O\left((\log n)^{1/3}\right)$ |
| Complete bipartite graphs | $O\left((\log n)^{1/3} \cdot T^{2/3}\right)$ $\Omega\left(T^{2/3}\right)$ | $O\left((\log n)^{\frac{1}{3}}\right)$ | $\Theta\left((\log n)^{1/3} \cdot T^{2/3}\right)$ | $O(1)$ |
| Orthogonal relation on $\mathbb{F}_2^k$ | $O\left((\log n)^{2/3} \cdot T^{2/3}\right)$ $\Omega\left(T^{2/3}\right)$ | $O\left((\log n)^{\frac{2}{3}}\right)$ | $\Theta\left((\log n)^{1/3} \cdot T^{2/3}\right)$ | $O(1)$ |

**Discussion.** *In general, our upper bound is never worse than the previous one since $\delta^* \le \delta$. Our lower bound is not directly comparable to the previously known lower bound as they are stated in terms of different parameters. In fact, we can not find an instance such that our lower bound $\Omega(\max\{1, (\delta^*/\alpha)^{1/3}\})$ is worse than the previous lower bound $\Omega(\max\{1, (\delta/(\log n)^2)^{1/3}\})$ and there are instances on which our bound outperforms. The two key quantities, namely the integrality gap $\frac{\delta}{\delta^*}$ of the primal linear programming and the integrality gap $\alpha$ of the dual linear programming, seem to be correlated in a graph. The relation between the two bounds is worth further investigation.*

**Related work**

The multi-armed bandit problem originated from the sequential decision making under uncertainty studied in [34, 6] and the adversarial bandit is a natural variant introduced by [7]. The work of [24] introduced the graph feedback model with a self-loop on each vertex in order to interpolate between the full feedback and bandit feedback settings. This model has been extensively studied in the work of [24, 4, 20, 3]. The work of [1] removed the self-loop assumption and considered generalized constrained graphs. They gave a full characterization of the mini-max regret in terms of the time horizon $T$. In contrast to fixed graph feedback, recent work of [20, 15, 2, 32] considered the time-varying graphs. Another line of recent work in [22, 23, 3] is to study random graphs, or the graphs with probabilistic feedback.

Most algorithms for adversarial bandits are derived from the EXP3 algorithm, e.g. [9, 29]. However, even for the vanilla multi-armed bandit problem, a direct application of EXP3 can only get an

upper bound of $O\left(\sqrt{n \log n \cdot T}\right)$ [9], which is suboptimal. In fact, the EXP3 is a special case of *online stochastic mirror descent* (OSMD) algorithm when using negentropy function as the potential function. OSMD was developed by [26] and [27] for online optimization. By choosing a more sophisticated potential function, namely the $1/2$-Tsallis entropy function [33], OSMD can achieve the tight bound $\Theta\left(\sqrt{nT}\right)$ [35].

The idea of using domination number or related parameters to study the feedback graph appeared many times in literature, e.g. [1, 12, 13, 4, 3]. The work of [3] used the idea of the fractional dominating set to study the high-probability regret bound for the strongly observable graph. Other similar works [23, 32, 13] mainly focused on stochastic settings. The follow-up works related to the weakly observable graph mainly considered harder settings including the time-varying graphs [2, 15, 3], bounded-memory adversaries [19] and the feedback graphs with switching costs [30, 5]. The recent work of [21] considered the bound with respect to cumulative losses of the best arm. To the best of our knowledge, there is no further development on the min-max regret since the work of [1].

## 2 Preliminaries

In this section, we formally describe the problem setting of bandits with graph feedback and introduce notations, definitions and propositions that will be used later.

Let $n \in \mathbb{N}$. We will use $[n]$ to denote the set $\{1, 2, \ldots, n\}$. Let $\mathbf{x} \in \mathbb{R}^n$ be an $n$-dimensional vector. For every $i \in [n]$, we use $\mathbf{x}(i)$ to denote the value on the $i$th-coordinate. We use $\{\mathbf{e}_1, \ldots, \mathbf{e}_n\}$ to denote the standard basis of $\mathbb{R}^n$. That is, $\mathbf{e}_i \in \mathbb{R}^n$ is the vector such that $\mathbf{e}_i(j) = \begin{cases} 1, & \text{if } j = i \\ 0, & \text{if } j \neq i \end{cases}$ for every $j \in [n]$. For every $n \in \mathbb{N}$, we define $\Delta_n \triangleq \left\{\mathbf{x} \in \mathbb{R}^n_{\geq 0} : \sum_{i=1}^n \mathbf{x}(i) = 1\right\}$ as the $n$-dimensional probability simplex. Clearly $\Delta_n$ is convex and every $\mathbf{x} \in \Delta_n$ can be viewed as a distribution on $[n]$. Throughout the article, sometimes we will view a function $\ell : [n] \to \mathbb{R}$ equivalently as a vector in $\mathbb{R}^n$, depending on which form is more convenient in the context. With this in mind, we have the inner product $\langle \ell, \mathbf{x} \rangle \triangleq \sum_{i \in [n]} \ell(i) \cdot \mathbf{x}(i)$ for every $\mathbf{x} \in \mathbb{R}^n$.

### 2.1 Graphs

In this article, we use $G = (V, E)$ to denote a directed graph with possible self-loops but no multiple edges. Therefore each $(u, v) \in E$ indicates a directed edge from $u$ to $v$ in $G$. If we say a graph $G = (V, E)$ is undirected, we view each undirected edge $\{u, v\} \in E$ as two directed edges $(u, v)$ and $(v, u)$. In the following, we assume $|V| \geq 2$ unless otherwise specified. For any $S \subseteq V$, $G[S]$ is the subgraph of $G$ induced by $S$. For every $v \in V$, we define $N_{\text{in}}(v) = \{u \in V : (u, v) \in E\}$ and $N_{\text{out}}(v) = \{u \in V : (v, u) \in E\}$ as the set of in-neighbors and out-neighbors of $v$ respectively. We also call $|N_{\text{in}}(v)|$ and $|N_{\text{out}}(v)|$ the in-degree and out-degree of $v$ respectively. A set $S \subseteq V$ is an *independent set* if there is no $u, v \in S$ such that $(u, v) \in E$. Note that we *do not* consider an isolated vertex with a self-loop as an independent set.

A vertex $v \in V$ is *strongly observable* if $(v, v) \in E$ or $\forall u \in V \setminus v, (u, v) \in E$. A vertex $v \in V$ is *non-observable* if $N_{\text{in}}(v) = \varnothing$. A directed graph $G$ is called *strongly observable* if each vertex of $G$ is strongly observable. It is called *non-observable* if it contains at least one non-observable vertex. The graph is called *weakly observable* if it is neither strong observable nor non-observable.

We say a directed graph $G$ is 1-degenerate if one can iteratively apply the following two operations in arbitrary orders on $G$ to get an empty graph: ● Pick a vertex with in-degree one and remove the in-edge; ● Pick a vertex with in-degree zero and out-degree at most one, and remove both the vertex and the out-edge. Typical 1-degenerate graphs include trees (directed or undirected) and directed cycles.

Let $U = \{i \in V : i \notin N_{\text{in}}(i)\}$ denote the set of vertices without self-loops. Consider the following linear program defined on $G$. We will call the linear program ($\mathscr{P}$).

$$\text{minimize} \sum_{i \in V} x_i; \quad \text{subject to} \sum_{i \in N_{\text{in}}(j)} x_i \geq 1, \ \forall j \in U; \quad 0 \leq x_i \leq 1, \ \forall i \in V. \qquad (\mathscr{P})$$

We use $\delta^*(G)$ to denote the optimum of the above linear program. We call $\delta^*(G)$ the *fractional weak domination number* of $G$. The dual of the linear program is

$$\text{maximize} \sum_{j \in U} y_j \quad \text{subject to} \sum_{j \in N_{\text{out}}(i) \cap U} y_j \leq 1, \ \forall i \in V; \quad 0 \leq y_j \leq 1, \ \forall j \in U. \quad (\mathscr{D})$$

We call this linear program ($\mathscr{D}$). We use $\zeta^*(G)$ to denote the optimum of the dual. We call $\zeta^*(G)$ the *fractional vertex packing number* of $G$. Then it follows from the *strong duality theorem* (see e.g. [11]) of linear programs that $\delta^*(G) = \zeta^*(G)$.

We remark that in [1], the weakly (integral) dominating set was defined to dominate all "weakly observable vertices" instead of "vertices without self-loops". These two definitions are all equivalent for all results in this article. See Appendix A for more explanations on this.

## 2.2 Bandits

Let $G = (V, E)$ be a directed graph where $V = [n]$ is the collection of bandit arms. Let $T \in \mathbb{N}$ be the time horizon which is known beforehand. The bandit problem is an online-decision game running for $T$ rounds. The player designs an algorithm $\mathscr{A}$ with the following behavior in each round $t$ of the game: • The algorithm $\mathscr{A}$ chooses an arm $A_t \in [n]$; • An adversary privately provides a loss function $\ell_t : \mathbb{N} \to [0, 1]$ and $\mathscr{A}$ pays a loss $\ell_t(A_t)$; • The algorithm receives feedback $\{\ell_t(j) : j \in N_{\text{out}}(A_t)\}$.

The *expected regret* of the algorithm $\mathscr{A}$ against a specific loss sequence $\ell^* = \{\ell_1, \ldots, \ell_T\}$ is defined by $R(G, T, \mathscr{A}, \ell^*) = \mathbf{E}\left[\sum_{t=1}^T \ell_t(A_t)\right] - \min_{a \in [n]} \sum_{t=1}^T \ell_t(a)$. Note that we look at the expectation of the algorithm since $\mathscr{A}$ might be randomized and it is not hard to see that randomization is necessary to guarantee $o(T)$ regret due to the adversarial nature of the loss sequence. The purpose of the problem is to design an algorithm performing well against the worst loss sequence, namely determining the min-max regret $R(G, T) \triangleq \inf_{\mathscr{A}} \sup_{\ell^*} R(G, T, \mathscr{A}, \ell^*)$.

There is another model called *stochastic bandits* in which the loss function at each round is not adversarially chosen but sampled from a fixed distribution. It is clear that this model is not harder than the one introduced above in the sense that any algorithm performing well in the adversarial setting also performs well in the stochastic setting. Therefore, we will construct instances of stochastic bandits to derive lower bounds in Section 4.

## 2.3 Optimization

Our upper bound is obtained via the online mirror descent algorithm. In this section, we collect a minimal set of terminologies to understand the algorithm. More details about the approach and its application to online decision-making can be found in e.g. [28].

Let $C \subseteq \mathbb{R}^n$. We use $\text{int}(C)$ to denote the interior $C$. For a convex function $\Psi : \mathbb{R}^d \to \mathbb{R} \cup \{\infty\}$, $\text{dom}(\Psi) \triangleq \{x : \Psi(x) < \infty\}$ is the domain of $\Psi$. Assume $\Psi$ is differentiable in its domain. For every $\mathbf{x}, \mathbf{y} \in \text{dom}(\Psi)$, $B_\Psi(\mathbf{x}, \mathbf{y}) \triangleq \Psi(\mathbf{x}) - \Psi(\mathbf{y}) - \langle \mathbf{x} - \mathbf{y}, \nabla\Psi(\mathbf{y}) \rangle \geq 0$ is the *Bregman divergence* between $\mathbf{x}$ and $\mathbf{y}$ with respect to the convex function $\Psi$. The diameter of $C$ with respect to $\Psi$ is $D_\Psi(C) \triangleq \sup_{\mathbf{x}, \mathbf{y} \in C}\{\Psi(\mathbf{x}) - \Psi(\mathbf{y})\}$. Let $A \in \mathbb{R}^{n \times n}$ be a *semi-definite positive* matrix and $\mathbf{x} \in \mathbb{R}^n$ be a vector. We define $\|\mathbf{x}\|_A \triangleq \sqrt{\mathbf{x}^\mathsf{T} A \mathbf{x}}$ as the norm of $\mathbf{x}$ with respect to $A$.

# 3 The algorithm

In this section, we design an algorithm to achieve the upper bound in Theorem 1. The proof is in Appendix B.

## 3.1 Online stochastic mirror descent (OSMD)

Our algorithm is based on the *Online Stochastic Mirror Descent* framework that has been widely used for bandit problems in various settings. Assuming the set of arms is $[n]$, possibly with many additional structures, a typical OSMD algorithm usually consists of following steps:

- Pick some initial distribution $X_1$ over all $n$ arms.
- For each round $t = 1, 2, \ldots, T$: – Tweak $X_t$ according to the problem structure to get a distribution $\tilde{X}_t$ over $n$ arms. – The adversary chooses some (unknown) loss vector $\ell_t : [n] \to [0, 1]$ with the knowledge of all previous information including $\tilde{X}_t$. The algorithm then picks an arm $A_t \sim \tilde{X}_t$ and pays a loss $\ell_t(A_t)$. After this, the algorithm observes some (partial) information $\Phi_t$ about $\ell_t$. – Construct an estimator $\hat{\ell}_t$ of $\ell_t$ using collected information $\Phi_t$, $A_t$ and $\tilde{X}_t$. – Compute an updated distribution $X_{t+1}$ from $X_t$ using mirror descent with a pre-specified potential function $\Psi$ and the estimated "gradient" $\hat{\ell}_t$.

Although the framework of OSMD is standard, there are several key ingredients left for the algorithm designer to specify: • How to construct the distribution $\tilde{X}_t$ from $X_t$? • How to construct the estimator $\hat{\ell}_t$? • How to pick the potential function $\Psi$? In general, filling these blanks heavily relies on the problem structure and sometimes requires ingenious construction to achieve low regret. We will first describe our algorithm and then explain our choices in Section 3.2.

### 3.2 The algorithm for bandits with graph feedback

Let $G = (V, E)$ be the input directed graph with $V = [n]$. A few offline preprocessing steps are required before the online part of the algorithm. We first solve the linear program ($\mathscr{P}$) to get an optimal solution $\{x_i^*\}_{i \in [n]}$. Recall $\delta^*(G) = \sum_{i \in [n]} x_i^*$ is the fractional weak domination number of $G$. Define a distribution $\mathbf{u} \in \Delta_n$ by normalizing $\{x_i^*\}_{i \in [n]}$, i.e., we let $\mathbf{u}(i) = \frac{x_i^*}{\sum_{j \in [n]} x_j^*}$ for all $i \in [n]$. The distribution $\mathbf{u}$ will be the *exploration distribution* whose function will be explained later. Define parameters $\gamma = \left( \frac{\delta^*(G) \log n}{T} \right)^{1/3}$, $\eta = \frac{\gamma^2}{\delta^*(G)}$ where $\gamma$ is the *exploration rate* and $\eta$ is the step size in the mirror descent. Finally, we let the potential function $\Psi : \mathbb{R}_{\geq 0}^n \to \mathbb{R}$ be $\mathbf{x} \in \mathbb{R}_{\geq 0}^n \mapsto \sum_{i=1}^n \mathbf{x}(i) \log \mathbf{x}(i)$ with the convention that $0 \cdot \log 0 = 0$. When restricted to $\Delta_n$, $\Psi(\mathbf{x})$ is the negative entropy of the distribution $\mathbf{x}$.

---

**Algorithm 1:** Online Stochastic Mirror Descent with Exploration

**begin**

    $X_1 \leftarrow \arg\min_{a \in \Delta_n} \Psi(a)$;

    **for** $t = 1, 2, \ldots, T$ **do**

        $\tilde{X}_t \leftarrow (1 - \gamma) \cdot X_t + \gamma \cdot \mathbf{u}$;

        /* use $\mathbf{u}$ to explore with rate $\gamma$.                                     */

        Play $A_t \sim \tilde{X}_t$ and observe $\ell_t(j)$ for all $j \in N_{\text{out}}(A_t)$;

        /* If $j \notin N_{\text{out}}(A_t)$, the value of $\ell_t(j)$ is unset.               */

        $\forall j \in [n]: \; \hat{\ell}_t(j) \leftarrow \frac{\mathbf{1}[j \in N_{\text{out}}(A_t)]}{\sum_{i \in N_{\text{in}}(j)} \tilde{X}_t(i)} \cdot \ell_t(j)$;

        /* For $j \notin N_{\text{out}}(A_t)$, $\hat{\ell}_t(j) = 0$.                            */

        $X_{t+1} \leftarrow \arg\min_{x \in \Delta_n} \eta \langle x, \hat{\ell}_t \rangle + B_\Psi(x, X_t)$;

        /* The update rule of mirror descent w.r.t. $\Psi$.                  */

    **end**

**end**

---

Clearly our algorithm implements OSMD framework by specializing the three ingredients mentioned in Section 3.1. • We choose $\tilde{X}_t = (1 - \gamma) \cdot X_t + \gamma \cdot \mathbf{u}$. This means that our algorithm basically follows $X_t$ but with a certain probability $\gamma$ to explore the arms according to $\mathbf{u}$. The reason for doing this is to guarantee that each arm has some not-so-small chance to be observed. It will be clear from the analysis of OSMD that the performance of the algorithm depends on the variance of $\hat{\ell}_t$, and a lower bound for each $\tilde{X}_t(i)$ implies an upper bound on the variance. On the other hand, we cannot choose $\gamma$ too large since it is $X_t$ who contains information on which arm is good, and our probability to follow $X_t$ is $1 - \gamma$. Therefore, our choice of $\gamma$ is optimized with respect to the trade-off between

the two effects. The Exp3.G algorithm in [1] used a uniform distribution over the weakly dominating set as an exploration probability instead of $\mathbf{u}$, which is the only difference between the two algorithms and leads to different graph parameters in regret bounds. Moreover, our exploration probability can be efficiently computed by solving the linear program $\mathscr{P}$ while it is NP-hard to determine theirs. • Our estimator $\hat{\ell}_t$ is a simple unbiased estimator for $\ell_t$, namely $\mathbf{E}\left[\hat{\ell}_t\right] = \ell_t$. • The potential function we used is the negative entropy function.

# 4 Lower bounds

In this section, we prove several lower bounds for the regret in terms of different graph parameters. All the lower bounds obtained in this section are based on a *meta lower bound* (Theorem 2) via the notion of $k$-*packing independent set*.

**Definition 6.** *Let $G(V, E)$ be a directed graph and $k \in \mathbb{N}$. A set $S \subseteq V$ is a $k$-packing independent set of $G$ if (1) $S$ is an independent set; (2) For any $v \in V$, it holds that $|N_{\text{out}}(v) \cap S| \leq k$.*

Intuitively, if a graph contains a large $k$-packing independent set $S$, then one can construct a hard instance as follows: • All arms in $V - S$ are bad, say with loss 1; • All arms in $S$ have loss $\text{Ber}\left(\frac{1}{2}\right)$ except a special one with loss $\text{Ber}\left(\frac{1}{2} - \varepsilon\right)$. Then any algorithm with low regret must successfully identify the special arm from $S$ without observing arms in $S$ much (since each observation of arms in $S$ comes from pulling $V - S$, which costs a penalty at least $\frac{1}{2}$ in the regret), and we can tweak the parameters to make this impossible. A similar idea already appeared in [1]. However, we will formally identify the problem of minimizing regret on this family of instances with the problem of *best arm identification*. Therefore, stronger lower bounds $\Omega\left(\max\left\{\frac{|S|}{k}, \log|S|\right\}^{\frac{1}{3}} \cdot T^{\frac{2}{3}}\right)$ in Theorem 2 can be obtained using tools from information theory.

**Remark.** *If the maximum independent set of the graph $G$ is of size one and $G$ is "weakly observable", then it has been shown in [1] that $R(G, T) = \Omega\left(T^{\frac{2}{3}}\right)$ for any algorithm. If the graph has no independent set, which means each vertex contains a self-loop, then the graph is "strongly observable" and its regret can be $O(T^{\frac{1}{2}})$. In particular, the problem of vanilla $n$-armed bandits falls into this case.*

We delay the proof of Theorem 2 to Appendix D and discuss some of its consequences in the remaining of this section. We recover and strengthen the lower bound based on the (integral) domination number of [1] in Section 4.1. Then we prove Theorem 3 in Section 4.2 and discuss some of its useful corollaries. Finally, we discuss in Section 4.3 when our lower bounds are optimal.

Our main technical contribution here is that we relate $\delta^*$ to the lower bound as well. This is achieved via applying the strong duality theorem of linear programming and using a new rounding method to construct hard instances from fractional solutions of the dual linear programming. This approach towards the lower bounds is much cleaner and in many cases stronger than previous ones in [1]. To the best of our knowledge, the method is new to the community of bandits algorithms

## 4.1 Lower bound via the (Integral) weak domination number

We first use Theorem 2 to recover and strengthen lower bounds in [1]. Let $G = (V, E)$ be a directed graph and $U \subseteq V$ be the set of vertices without self-loops.

The weakly dominating set of $U$ is a set $S \subseteq V$ such that for every $u \in U$, there exists some $v \in S$ with $(v, u) \in E$. The weak domination number, denoted by $\delta(G)$, is the size of the minimum weakly dominating set of $U$. In fact, $\delta(G)$ is the optimum of the integral restriction of the linear program ($\mathscr{P}$) in Section 2.1:

$$\text{minimize} \sum_{i \in V} x_i; \quad \text{subject to} \sum_{i \in N_{\text{in}}(j)} x_i \geq 1, \ \forall j \in U; \quad x_i \in \{0, 1\}, \ \forall i \in V. \qquad (\mathscr{P}')$$

The following structural lemma was proved in [1]

**Lemma 7.** *The graph $G$ contains a $(\log n)$-packing independent set $S$ of size at least $\frac{\delta(G)}{50 \log n}$.*

Applying Lemma 7 to Theorem 2, we obtain

**Theorem 8.** *If $G$ is weakly observable, then for any algorithm and any sufficiently large time horizon $T \in \mathbb{N}$, there exists a sequence of loss functions such that $R(G,T) = \Omega\left(\max\left\{\frac{\delta(G)}{\log^2 n}, \log\left(\frac{\delta(G)}{\log n}\right)\right\}^{1/3} \cdot T^{2/3}\right).$*

Note that the bound in [1] is $R(G,T) = \Omega\left(\max\left\{\frac{\delta(G)}{\log^2 n}, 1\right\}^{1/3} \cdot T^{2/3}\right)$. Theorem 8 outperforms when $\omega(\log n) < \delta(G) < o(\log^2 n \cdot \log \log n)$.

## 4.2 Lower bound via the linear program dual

In this section, we use Theorem 2 to derive lower bounds in terms of $\delta^*(G)$. Recall the linear program ($\mathscr{D}$) in Section 2.1 whose optimum is $\zeta^*(G) = \delta^*(G)$ by the strong duality theorem of linear programming. Consider its integral restriction ($\mathscr{D}'$):

$$\text{maximize} \sum_{j \in U} y_j \quad \text{subject to} \sum_{j \in N_{\text{out}}(i) \cap U} y_j \leq 1, \ \forall i \in V; \quad y_j \in \{0,1\}, \ \forall j \in U. \qquad (\mathscr{D}')$$

For every feasible solution $\{\hat{y}_j\}_{j \in U}$ of ($\mathscr{D}'$), the set $S \triangleq \{j \in U : \hat{y}_j = 1\}$ is called a *vertex packing set* on $U$. It enjoys the property that for every $i \in V$, $|N_{\text{out}}(i) \cap S| \leq 1$.

Let $\zeta(G)$ be the optimum of ($\mathscr{D}'$), namely the size of the maximum vertex packing set on $U$. Let $\alpha \triangleq \frac{\zeta^*(G)}{\zeta(G)}$ be the integrality gap of ($\mathscr{D}$). In the following, we will write $\delta^*, \delta, \zeta^*, \zeta$ instead of $\delta^*(G), \delta(G), \zeta^*(G), \zeta(G)$ respectively if the graph $G$ is clear from the context.

We can use a greedy algorithm to find an 1-packing independent set of size at least $\frac{|S|}{3}$ in $S$ and then Theorem 3 follows from Theorem 2. Theorem 3 is less informative if we do not know how large the integrality gap $\alpha$ is. On the other hand, the integrality gap of linear programs for packing programs has been well-studied in the literature of approximation algorithms. The following bound due to [31] is tight for general graphs.

**Lemma 9 ([31]).** *For any directed graph $G$, the integrality gap $\alpha = O\left(n/\delta^*\right)$.*

**Corollary 10.** *If $G$ is weakly observable, then for any algorithm and any sufficiently large time horizon $T \in \mathbb{N}$, there exists a sequence of loss functions such that $R(G,T) = \Omega\left(\left(\frac{\delta^{*2}}{n}\right)^{\frac{1}{3}} \cdot T^{\frac{2}{3}}\right).$*

The bound for the integrality gap in Lemma 9 holds for any graphs. It can be greatly improved for special graphs.

An interesting family of graphs with small $\alpha$ is those bounded degree graphs. If the in-degree of every vertex in $U$ is bounded, we have the following bound for the integrality gap:

**Lemma 11 ([10]).** *If the in-degree of every vertex in $U$ is bounded by a constant $\Delta$, then the integrality gap $\alpha \leq 8\Delta$.*

**Corollary 12.** *Let $\Delta \in \mathbb{N}$ be a constant and $\mathcal{G}_\Delta$ be the family of graphs with maximum in-degree $\Delta$. Then for every weakly observable $G = (V, E) \in \mathcal{G}_\Delta$, any algorithm and any sufficiently large time horizon $T \in \mathbb{N}$, there exists a sequence of loss functions such that $R(G,T) = \Omega((\delta^*)^{\frac{1}{3}} \cdot T^{\frac{2}{3}})$.*

Next, we show the integrality gap of another broad family of graphs, the 1-degenerate graphs, is 1. The family of 1-degenerate graphs was defined in Section 2.1. Graphs including trees (both directed and undirected), directed cycles belong to this family. The proof of the following lemma is in Appendix C.2.

**Lemma 13.** *For any 1-degenerate directed graph, the integrality gap $\alpha = 1$.*

**Corollary 14.** *Let $G$ be a 1-degenerate weakly observable graph. Then for any algorithm and any sufficiently large time horizon $T \in \mathbb{N}$, there exists a sequence of loss functions such that $R(G,T) = \Omega((\delta^*)^{\frac{1}{3}} \cdot T^{\frac{2}{3}})$.*

We obtained in Theorem 1 that $R(G,T) = O\left((\delta^* \log n)^{\frac{1}{3}} T^{\frac{2}{3}}\right)$, and therefore our lower bound is tight up to a factor of $(\log n)^{\frac{1}{3}}$ on 1-degenerate graphs and graphs of bounded degree.

### 4.3 Instances with optimal regret

In this section, we will examine several families of graphs in which the optimal regret bound can be obtained using tools developed in this article.

#### 4.3.1 Complete bipartite graphs

Let $G = (V_1 \cup V_2, E)$ be an undirected complete bipartite graph with $n = |V_1| + |V_2|$. Clearly $\delta^* = \delta = 2$. Therefore both our Theorem 1 and the algorithm in [1] satisfy $R(G,T) = O\left((\log n)^{\frac{1}{3}} \cdot T^{\frac{2}{3}}\right)$.

Assuming without loss of generality that $|V_1| \geq |V_2|$, then $V_1$ is a $|V_1|$-packing independent set of size at least $\frac{n}{2}$. Therefore it follows from Theorem 2 that any algorithm satisfies $R(G,T) = \Omega\left((\log n)^{\frac{1}{3}} \cdot T^{\frac{2}{3}}\right)$. Note that the lower bound in [1] is $\Omega\left(T^{\frac{2}{3}}\right)$ for this instance.

#### 4.3.2 Orthogonal relation on $\mathbb{F}_2^k$

Our algorithm outperforms the one in [1] when $\delta^* \ll \delta$. Let us now construct a family of graphs where $\frac{\delta}{\delta^*} = \Omega(\log n)$.

Let $\mathbb{F}_2$ be the finite field with two elements and $k \in \mathbb{N}$ be sufficiently large. The vertex set of the undirected graph $G = (V_1 \cup V_2, E)$ consists of two disjoint parts $V_1$ and $V_2$ where $V_1$ and $V_2$ are both isomorphic to $\mathbb{F}_2^k \setminus \{\mathbf{0}\}$. Therefore we can write $V_1 = \{x_\alpha\}_{\alpha \in \mathbb{F}_2^k \setminus \{\mathbf{0}\}}$ and $V_2 = \{y_\beta\}_{\beta \in \mathbb{F}_2^k \setminus \{\mathbf{0}\}}$. The set of edges $E$ is as follows: $\bullet$ $E$ is the edge set such that $G[V_1]$ is a clique and $G[V_2]$ is an independent set; $\bullet$ For every $x_\alpha \in V_1$ and $y_\beta \in V_2$, $\{x_\alpha, y_\beta\} \in E$ iff $\langle \alpha, \beta \rangle = \sum_{i=1}^{k} \alpha(i) \cdot \beta(i) = 1$, where all multiplications and summations are in $\mathbb{F}_2$.

We will show that the upper bound and the lower bound of the regret for this instance proved in [1] based on $\delta$ is $O\left((\log n)^{\frac{2}{3}} \cdot T^{\frac{2}{3}}\right)$ and $\Omega\left(T^{\frac{2}{3}}\right)$ respectively. However, we achieve the optimal $\Theta\left((\log n)^{\frac{1}{3}} \cdot T^{\frac{2}{3}}\right)$ regret. Thus we conclude that both our new upper bound and lower bound are crucial for the optimal regret on this family of instances. The details can be found in Appendix C.3

## 5 Conclusion

In this article, we introduced the notions of fractional weak domination number and $k$-packing independence number respectively to prove new upper bounds and lower bounds for the regret of bandits with graph feedback. Our results implied optimal regret on several families of graphs. Although there are still some gaps in general, we believe that these two notions are the correct graph parameters to characterize the complexity of the problem. We now list a few interesting problems worth future investigation.

- Let $G$ be an $n$-vertex undirected cycle. What is the regret on this instance? Theorem 1 implies an upper bound $O\left((n \log n)^{\frac{1}{3}} T^{\frac{2}{3}}\right)$ and Theorem 2 implies a lower bound $\Omega\left(n^{\frac{1}{3}} T^{\frac{2}{3}}\right)$.

- The lower bound $\Omega\left(\left(\frac{\delta^*}{\alpha}\right)^{\frac{1}{3}} \cdot T^{\frac{2}{3}}\right)$ in Theorem 3 for general graphs is not satisfactory. The lower bound is proved via the construction of a 1-packing independent set. This construction did not release the full power of Theorem 2 as the lower bound in the theorem applies for any $k$-packing independent set. It is still possible to construct larger $k$-packing independent sets via rounding the linear program $\mathscr{D}$ to some "less integral" solution.

- Is Theorem 2 tight? In fact, the bound for BESTARMID, which is constructed to prove Theorem 2 in the full version of the paper, is tight since matching upper bound exists. Therefore, one needs new constructions of hard instances to improve Theorem 2, if possible.

## Acknowledgements

This work was partly supported by National Key Research and Development Program of China (2020AAA0107600), National Natural Science Foundation of China Grant (62006151, 62076161,61802069,61902241), Shanghai Sailing Program, and Shanghai Science and Technology Commission Grant (17JC1420200).

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
