# A Two definitions of weak dominating set

In [1], the weakly (integral) dominating set was defined to dominate all "weakly observable vertices" instead of "vertices without self-loops". This was indeed a flaw in the paper as in some extreme cases, the set may fail to dominate vertices in $U$ (the set of vertices without self-loops) that are "strongly observable". Therefore we ask for the set to dominate $U$. Nevertheless the two definitions of domination number, both integral and fractional, differ by at most one and do not affect the asymptotic bounds.

Here we give an example to explain the difference between the definition of the weak domination number $\delta$ in [1] and in this article. To ease the presentation, we call them $\delta$ and $\delta'$ respectively. We show that $\delta = \delta'$ if $\delta \geq 2$, but it is possible that $\delta = 1$ and $\delta' = 2$.

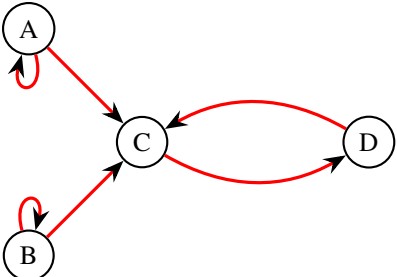

Figure 1: An example of $\delta = 1$, $\delta' = 2$

It is clear that $\delta = 1$ and $\delta' = 2$ in Figure 1 since the vertex $C$ is strongly observable and in [1], the vertex $C$ does not need to be dominated by a weak dominating set. Therefore, the minimum weak dominating set is $\{C\}$. However, in the proof of [1, Theorem 2] for the weakly observable graphs, they assumed that every vertex without a self-loop is dominated by the weakly dominating set. This is not true following their definition since the vertex $C$, although strongly observable, is not dominated by itself and thus the lower bound on the probability that $C$ is observed fails.

Hence we ask for the set to dominate the set of vertices without self-loops, namely $U = \{C, D\}$. The proof can then go through, and the only difference is that $\delta'$ becomes to two. It is also clear when $\delta \geq 2$, this situation will not occur as every strongly observable vertex without a self-loop can be dominated by any vertex other than itself.

# B Proof of Theorem 1

Since our algorithm only deviates from the standard OSMD algorithm by incorporating an additional exploration term $\gamma \cdot \mathbf{u}$, the regret naturally consists of two parts: the standard OSMD regret and the amount introduced by the additional exploration.

Fix a sequence of loss functions $\ell_1, \ldots, \ell_T$ and let $a^* = \arg\min_{a \in [n]} \sum_{t=1}^{T} \ell_t(a)$ be the optimal arm.

**Lemma 15.** *For any time horizon $T \in \mathbb{N}$, the Algorithm 1 satisfies*

$$R(G, T) \leq \sum_{t=1}^{T} \mathbf{E}\left[ \langle X_t - \mathbf{e}_{a^*}, \hat{\ell}_t \rangle \right] + \gamma T .$$

*Proof.* For every $t = 1, 2, \ldots, T$, let $\mathcal{F}_t$ be the $\sigma$-algebra generated by the random variables appeared at and before the $t$-th round. Define $\mathbf{E}_t[\cdot] = \mathbf{E}[\cdot | \mathcal{F}_t]$, then

$$R(G, T) = \mathbf{E}\left[ \sum_{t=1}^{T} \ell_t(A_t) \right] - \sum_{t=1}^{T} \ell_t(a^*) = \sum_{t=1}^{T} \mathbf{E}\left[ \mathbf{E}_{t-1}\left[ \ell_t(A_t) \right] \right] - \sum_{t=1}^{T} \mathbf{E}\left[ \ell_t(a^*) \right] .$$

Since $\tilde{X}_t$ is $\mathcal{F}_{t-1}$-measurable and $A_t \sim \tilde{X}_t$, we have

$$
\begin{aligned}
R(G,T) &= \sum_{t=1}^{T} \mathbf{E}\left[\mathbf{E}_{t-1}\left[\ell_t(A_t)\right]\right] - \sum_{t=1}^{T} \mathbf{E}\left[\ell_t(a^*)\right] \\
&= \sum_{t=1}^{T} \mathbf{E}\left[\langle \tilde{X}_t, \ell_t \rangle\right] - \sum_{t=1}^{T} \langle \mathbf{e}_{a^*}, \ell_t \rangle \leq \sum_{t=1}^{T} \mathbf{E}\left[\langle X_t + \gamma \cdot \mathbf{u}, \ell_t \rangle\right] - \sum_{t=1}^{T} \langle \mathbf{e}_{a^*}, \ell_t \rangle \\
&= \sum_{t=1}^{T} \mathbf{E}\left[\langle X_t - \mathbf{e}_{a^*}, \ell_t \rangle\right] + \sum_{t=1}^{T} \gamma \cdot \langle \mathbf{u}, \ell_t \rangle \leq \sum_{t=1}^{T} \mathbf{E}\left[\langle X_t - \mathbf{e}_{a^*}, \hat{\ell}_t \rangle\right] + \gamma \cdot T,
\end{aligned}
$$

where in the last inequality we used the facts that $\hat{\ell}_t$ is an unbiased estimator for $\ell_t$ and $\langle \mathbf{u}, \ell_t \rangle \leq 1$.∎

Expanding the first term, we have the following result.

**Lemma 16.**

$$
R(G,T) \leq \frac{D_\Psi(\Delta_n)}{\eta} + \frac{\eta}{2} \sum_{t=1}^{T} \mathbf{E}_{A_t \sim \tilde{X}_t}\left[\sup_{\mathbf{z} \in [Y_t, X_t]} \left\|\hat{\ell}_t\right\|^2_{(\nabla^2\Psi(\mathbf{z}))^{-1}}\right] + \gamma T,
$$

*where $Y_t = \arg\min_{\mathbf{y} \in \mathrm{int}(\mathrm{dom}(\Psi))} \left(\eta\langle \mathbf{y}, \hat{\ell}_t \rangle + B_\Psi(\mathbf{y}, X_t)\right)$.*

Lemma 16 is a consequence of Lemma 15 and an upper bound for $\sum_{t=1}^{T} \mathbf{E}\left[\langle X_t - \mathbf{e}_a, \hat{\ell}_t \rangle\right]$. The latter is a standard regret bound for OSMD and a proof can be found in, e.g. [35] and references therein.

We are now ready to prove Theorem 1. It is proved by plugging our choice of potential function into the bound of Lemma 16.

**Theorem 1.** *There exists an algorithm such that for any weakly observable graph, any time horizon $T \geq n^3 \log(n)/\delta^{*2}(G)$, its regret satisfies $R(G,T) = O\left((\delta^*(G)\log n)^{\frac{1}{3}} T^{\frac{2}{3}}\right)$.*

*Proof.* Recall that we choose $\Psi(\mathbf{x}) = \sum_{i \in [n]} \mathbf{x}(i) \log \mathbf{x}(i)$ for every $\mathbf{x} \in \Delta_n$. Direct calculation yields $D_\Psi(\Delta_n) \leq \log n$, $(\nabla^2\Psi(\mathbf{z}))^{-1} = \mathrm{diag}(\mathbf{z}(1), \ldots, \mathbf{z}(n))$ and $Y_t(i) = X_t(i) \cdot e^{-\eta\hat{\ell}_t(i)} \leq X_t(i)$ for all $t$ and $i \in [n]$. Therefore

$$
\begin{aligned}
\mathbf{E}_{A_t \sim \tilde{X}_t}\left[\sup_{\mathbf{z} \in [X_t, Y_t]} \left\|\hat{\ell}_t\right\|^2_{(\nabla^2\psi(\mathbf{z}))^{-1}}\right] &= \mathbf{E}_{A_t \sim \tilde{X}_t}\left[\sup_{\mathbf{z} \in [X_t, Y_t]} \sum_{i=1}^{n} \frac{\mathbf{1}\left[i \in N_{\mathrm{out}}(A_t)\right]^2}{\left(\sum_{j \in N_{\mathrm{in}}(i)} \tilde{X}_t(j)\right)^2} \cdot \ell_t(i)^2 \cdot \mathbf{z}(i)\right] \\
&\leq \mathbf{E}\left[\sum_{i=1}^{n} \frac{X_t(i)}{\sum_{j \in N_{\mathrm{in}}(i)} \tilde{X}_t(j)}\right].
\end{aligned}
$$

It remains to lower bound $\sum_{j \in N_{\mathrm{in}}(i)} \tilde{X}_t(j)$ for every $i \in [n]$, which is the probability that the arm $i$ is observed at the $t$-th round. We require that the probability is not too small compared to $X_t(i)$ for every $i \in [n]$. Recall $U = \{i \notin N_{\mathrm{in}}(i)\}$ denotes the set of vertices without self-loops. Then for every $i \notin U$, the self-loop on $i$ guarantees that the chance for $i$ to be observed is comparable to $X_t(i)$. On the other hand, if $i \in U$, we use our additional exploration term $\gamma \cdot \mathbf{u}$ to lower bound the probability. It is clear that $\gamma \leq \frac{1}{2}$ by our choice of $\gamma$ and $T$. So the contribution of vertices in $V \setminus U$ is

$$
\sum_{i \notin U} \frac{X_t(i)}{\sum_{j \in N_{\mathrm{in}}(i)} \tilde{X}_t(j)} = \sum_{i \notin U} \frac{X_t(i)}{\sum_{j \in N_{\mathrm{in}}(i)} \left((1-\gamma) \cdot X_t(j) + \gamma \cdot \mathbf{u}(j)\right)} \leq \sum_{i \notin U} \frac{1}{1-\gamma} \leq 2n. \quad (1)
$$

The contribution of vertices in $U$ is

$$
\sum_{i \in U} \frac{X_t(i)}{\sum_{j \in N_{\mathrm{in}}(i)} \tilde{X}_t(j)} \leq \sum_{i \in U} \frac{X_t(i)}{\gamma \sum_{j \in N_{\mathrm{in}}(i)} \mathbf{u}(j)} \overset{(\heartsuit)}{\leq} \frac{\sum_{i \in U} X_t(i) \cdot \delta^*(G)}{\gamma} \leq \frac{\delta^*(G)}{\gamma}, \quad (2)
$$

where ($\heartsuit$) is due to the first constraint of the linear program $\mathscr{P}$ and our definition of $\mathbf{u}$.

Combining (1) and (2), we obtain

$$R(G, T) \leq \frac{\log n}{\eta} + \eta n T + \frac{\eta \delta^*(G) T}{2\gamma} + \gamma T. \tag{3}$$

The theorem follows by plugging in our parameters $\gamma = \left(\frac{\delta^*(G) \log n}{T}\right)^{1/3}$, $\eta = \frac{\gamma^2}{\delta^*(G)}$ and assuming $T \geq n^3 \log n / \delta^*(G)^2$. ∎

## C   Proof of lower bounds

### C.1   Proof of Theorem 3

**Theorem 3.** *If $G$ is weakly observable, then for any algorithm and any sufficiently large time horizon $T \in \mathbb{N}$, there exists a sequence of loss functions such that $R(G, T) = \Omega\left(\left(\frac{\delta^*}{\alpha}\right)^{1/3} \cdot T^{\frac{2}{3}}\right)$, where $\alpha$ is the integrality gap of the linear program for vertex packing.*

*Proof.* Since $\delta^* = \zeta^* = \alpha \cdot \zeta$, the bound in Theorem 3 is equivalent to $R(G, T) = \Omega\left(\zeta^{\frac{1}{3}} \cdot T^{\frac{2}{3}}\right)$. We will prove that $G$ contains a 1-packing independent set of size $\Theta(\zeta)$, then the theorem follows from Theorem 2.

Let $\left\{y_j^\dagger\right\}_{j \in U}$ be the optimal solution of the integral program $\mathscr{D}'$. Let $S^\dagger \triangleq \left\{j \in U : y_j^\dagger = 1\right\}$ be the corresponding vertex packing set. Then clearly $\zeta = |S^\dagger|$. We will show that there exists a 1-packing independent set $H \subseteq S^\dagger$ with $|H| \geq |S^\dagger|/3$.

We use the following greedy strategy to construct $H$.

- INITIALIZATION: Set $H = \emptyset$ and $S' = S^\dagger$.

- UPDATE: While $S' \neq \emptyset$: Pick a vertex $v \in S'$ with minimum $|N_{\text{in}}(v) \cap S'|$; Set $H \leftarrow H \cup \{v\}$; $S' \leftarrow S' \setminus (N_{\text{in}}(v) \cup \{v\} \cup N_{\text{out}}(v))$.

First of all, the set $H$ constructed above must be an independent set as whenever we add some vertex $v$ into $H$, we remove all its incident vertices, both its in-neighbors and out-neighbors, from $S'$. Clearly each $S'$ after removing these vertices is still a vertex packing set. Therefore, every $v \in S'$ has out-degree at most one in $G[S']$. This implies that the vertex $v \in S'$ with minimum $|N_{\text{in}}(v) \cap S'|$, or equivalently minimum in-degree in $G[S']$, satisfies $|N_{\text{in}}(v) \cap S'| \leq 1$. So the update step $S' \leftarrow S' \setminus (N_{\text{in}}(v) \cup \{v\} \cup N_{\text{out}}(v))$ removes at most three vertices from $S'$. This concludes that $H$ is a 1-packing independent set of size at least $|S^\dagger|/3$. ∎

### C.2   Proof of Lemma 13

**Lemma 13.** *For any 1-degenerate directed graph, the integrality gap $\alpha = 1$.*

*Proof.* Let $G = (V, E)$ be a 1-degenerate directed graph. We show that we can construct a vertex packing set $S$ with $|S| \geq \sum_{j \in U} \hat{y}_j$ for any feasible solution $\{\hat{y}_j\}_{j \in U}$ of $\mathscr{D}$. The lemma follows by applying this to the optimal solution.

We use $\mathbf{y}_S \in \{0, 1\}^U$ to denote the indicator vector of $S$. So we have for every $i \in U$, $\mathbf{y}_S(i) = 1 \iff i \in S$. The construction is to apply the following greedy strategy to determining $\mathbf{y}_S$ until the graph is empty:

- Pick a vertex $i$ in $G$ with in-degree one. Let $(j, i)$ be the unique in-edge of $i$. Remove the edge $(j, i)$ from $E$. If $i \in U$ and the value of $\mathbf{y}_S(i)$ is not determined, then (1) set $\mathbf{y}_S(i) = 1$; (2) for all $k \in U \setminus \{i\}$ such $(j, k) \in E$, set $y_S(k) = 0$.

- Pick a vertex with in-degree zero and out-degree at most one, and remove both the vertex and the out-edge. Keep doing so until no such vertex exists in $G$.

It is clear that the above algorithm terminates at an empty $G$ since all operations to the graph coincide with ones defining 1-degeneration. We only need to verify that

1. $\mathbf{y}_S$ is a feasible solution of $\mathscr{D}$; and

2. $\sum_{j \in U} \mathbf{y}_S(j) \geq \sum_{j \in U} \hat{y}_j$ for any feasible solution $\{\hat{y}_j\}_{j \in U}$.

Let us first verify (1). The vector $\mathbf{y}_S$ can become infeasible only when we set some $\mathbf{y}_S(i) = 1$. Note that this happens only when the in-degree of $i$ is one, or equivalent there is only a unique edge $(j, i)$ pointing to $i$. We do not violate the constraint on $j$ as we set all $y_S(k) = 0$ for $k \in U \setminus \{i\}$ and $(j, k) \in E$. It is still possible that there exists some other $j' \in V$ such that the edge $(j', i)$ exists but has been removed. Since the value of $\mathbf{y}_S(i)$ is not determined before, this happens only if $j'$ has out-degree one at the beginning, and so the constraint on $j'$ cannot be violated either.

To see (2), we assume the value on each $j \in U$ is $\hat{y}_j$ at the beginning. Our procedure to construct $\mathbf{y}_S$ can be equivalently viewed as a process to change each $\hat{y}_j$ to either 0 or 1. That is, after we set the value of some $\mathbf{y}_S(j)$ to 0 or 1, we change $\hat{y}_j$ to the same value. It is easy to verify that during the process, we never decrease $\sum_{j \in U} \hat{y}_j$. At last, $\mathbf{y}_S(j) = \hat{y}_j$ for all $j \in U$ and some optimal solution $\{\hat{y}_j\}_{j \in U}$, and (2) is verified. ∎

### C.3 Proof for special graphs in Section 4.3.2

Let $n = |V_1| + |V_2| = 2^{k+1} - 2$. It is clear that the degree of each vertex $y_\beta \in V_2$ is $2^{k-1} = \frac{n+2}{4}$. A moment's reflection should convince you that $\delta^*(G) \leq 2$ as we can put a fraction of $\frac{4}{n+2}$ on each $x_\alpha \in V_1$.

Therefore it follows from Theorem 1 that our algorithm has regret $O\left((\log n)^{\frac{1}{3}} \cdot T^{\frac{2}{3}}\right)$ on this family of instances. Moreover, $V_2$ is a $\frac{n+2}{4}$-packing independent set of size $\frac{n}{2}$. It follows from Theorem 2 that any algorithm has regret $\Omega\left((\log n)^{\frac{1}{3}} \cdot T^{\frac{2}{3}}\right)$.

Finally, we remark that $\delta(G) = k = \log_2\left(\frac{n+2}{2}\right)$. To see this, we first observe that the minimum dominating set of the graph must reside in $V_1$ since $G[V_2]$ is an independent set. Then we show any $S \subseteq V_1$ with $|S| \leq k - 1$ cannot dominate all vertices in $V_2$. Assume $S = \{x_{\alpha_1}, x_{\alpha_2}, \ldots, x_{\alpha_{k-1}}\}$. In fact, a vertex $y_\beta \in V_2$ is dominated by $S$ iff $\langle \alpha_i, \beta \rangle = 1$ for some $i \in [k-1]$. In other words, if we view each $\alpha_i$ as a column vector in $\mathbb{F}_2^k$ and define the matrix $A = [\alpha_1, \alpha_2, \ldots, \alpha_{k-1}]$, then $y_\beta$ is dominated by $S$ iff $A^\mathsf{T}\beta \neq \mathbf{0}$. However, the dimension of $A^\mathsf{T}$ is at most $k - 1$ and therefore by the rank-nullity theorem that its null space is of dimension at least one. This means that there exists a certain $\beta' \in V_2$ with $A^\mathsf{T}\beta' = \mathbf{0}$. So $y_{\beta'}$ is not dominated.

The above fact implies that the upper bound and the lower bound of the regret for this instance proved in [1] based on $\delta$ are $O\left((\log n)^{\frac{2}{3}} \cdot T^{\frac{2}{3}}\right)$ and $\Omega\left(T^{\frac{2}{3}}\right)$ respectively.

## D  Proof of Theorem 2

**Theorem 2.** *Let $G = (V, E)$ be a directed graph. If $G$ contains a $k$-packing independent set $S$ with $|S| \geq 2$, then for any randomized algorithm and any time horizon $T$, there exists a sequence of loss functions such that the expected regret is $\Omega\left(\max\left\{\frac{|S|}{k}, \log|S|\right\}^{\frac{1}{3}} \cdot T^{\frac{2}{3}}\right)$.*

Our strategy to prove Theorem 2 is to reduce the problem of minimizing regret to the problem of *best arm identification*. We first define the problem and discuss its complexity in Appendix D.2 with the help of Appendix D.1. Then we construct the reduction and prove Theorem 2 in Appendix D.3.

### D.1  Information Theory

We borrow tools from information theory to establish lower bounds. More details can be found in the standard textbook [16] on the topic. To ease the notation, each "log" appeared in the article

without subscript is of base $e$. We fix a probability space $(\Omega, \mathcal{F}, \mathbf{Pr})$ and let $X, Y : \Omega \to U$ be discrete-valued random variables for a finite set $U$.

The entropy of $X$ is $H(X) \triangleq - \sum_{x \in U} \mathbf{Pr}[X = x] \cdot \log \mathbf{Pr}[X = x]$. The conditional information $H(X|Y) \triangleq - \sum_{x,y \in U} \mathbf{Pr}[X = x, Y = y] \log \mathbf{Pr}[X = x \mid Y = y]$. The mutual information between $X$ and $Y$ is $I(X;Y) \triangleq \sum_{x,y \in U} \mathbf{Pr}[X = x, Y = y] \cdot \log \frac{\mathbf{Pr}[X=x,Y=y]}{\mathbf{Pr}[X=x] \cdot \mathbf{Pr}[Y=y]}$. It is a basic fact that $H(X) = H(X|Y) + I(X;Y)$. Suppose we have another random variable $Z$, then $I(X;Y|Z) \triangleq H(X|Z) - H(X|Y, Z)$.

Suppose $Z : \Omega \to W$ is a random variable correlated to $X$ and one needs to guess the value of $X$ via observing $Z$. The Fano's inequality reveals the inherent difficulty of this task:

**Lemma 17 (Fano's inequality [18]).** *For any function $f : W \to U$, it holds that*

$$\mathbf{Pr}[f(Z) \neq X] \geq \frac{H(X) - I(X;Z) - \log 2}{\log |U|}.$$

If we assume $Y = (Y_1, \ldots, Y_n)$ is a vector of random variables such that $\{Y_i\}_{i \in [n]}$ are mutually independent conditional on $X$, then we have the following lemma of tensorization of mutual information:

**Lemma 18 (Tensorization of Mutual Information).** *If $Y = (Y_1, \ldots, Y_n)$ and random variables $\{Y_i\}_{i \in [n]}$ are mutually independent conditional on $X$, then*

$$I(X;Y) \leq \sum_{i=1}^{n} I(X;Y_i).$$

*Proof.* By the chain rule of the mutual information,

$$I(X;Y) = \sum_{i=1}^{n} I(X;Y_i|Y_1, \ldots, Y_{i-1}).$$

For every $i \in [n]$, we have

$$I(X;Y_i|Y_1, \ldots, Y_{i-1}) = H(Y_i|Y_1, \ldots, Y_{i-1}) - H(Y_i|X, Y_1, \ldots, Y_{i-1}) \leq H(Y_i) - H(Y_i|X) = I(X;Y_i),$$

where we use the fact that $H(Y_i|X, Y_1, \ldots, Y_{i-1}) = H(Y_i|X)$ due to the conditional mutual independence. $\blacksquare$

### D.2 Best arm identification

The problem of *best arm identification* is formally defined as follows.

---

BEST ARM IDENTIFICATION (BESTARMID)
    *Input:*    An instance of $n$ stochastic arms where the loss of the $i$-th arm follows $\mathrm{Ber}(p_i)$. Each pull of arm $i$ receives a loss $\sim \mathrm{Ber}(p_i)$ independently.
    *Problem:*  Determine the arm $i$ with minimum $p_i$ via pulling arms.

---

Therefore, an instance of BESTARMID is specified by a vector $\mathbf{p} = (p_1, \ldots, p_n) \in [0,1]^n$. The goal is to design a strategy to find the arm $i$ with minimum $p_i$ via pulling arms. We call the arm with minimum $p_i$ the best arm. We want to minimize the number of pulls in total and the main result of this section is to provide lower bounds for this task: For some collection of vectors $\mathbf{p}$, if the total number of pulls is below some threshold, then any algorithm cannot find the best arm for all instances with high probability.

In the following, we may abuse notations and simply say a vector $\mathbf{p} \in [0,1]^n$ is an instance of BESTARMID. Now for every $j \in [n]$, we define an instance $\mathbf{p}^{(j)} = (p_1^{(j)}, \ldots, p_n^{(j)}) \in \mathbb{R}^n$ as

$$p_i^{(j)} = \begin{cases} \frac{1}{2} - \varepsilon, & j = i; \\ \frac{1}{2}, & j \neq i, \end{cases} \text{ for some } \varepsilon \in (0, \tfrac{1}{2}].$$ This is the collection of instances we will study. For the convenience of the reduction in Lemma 19, we also define $\mathbf{p}^{(0)}$ as an $n$-dimensional vector of all $\frac{1}{2}$.

There are several ways to explore the $n$ arms in order to determine the one with minimum mean. We first consider the most general strategy: In each round, the player can pick an arbitrary subset $S \subseteq [n]$ and pull the arms therein. The game proceeds for $T$ rounds and then the player needs to determine the best arm with collected information. Note that in each round, the exploring set $S$ may adaptively depend on previous information.

Now we fix such a (possibly randomized) strategy and denote it by $\mathscr{B}$. For every $j \in [n]$ and $i \in [n]$, we use $N_i^{(j)}$ to denote the number of times that the $i$-th arm is pulled when we run $\mathscr{B}$ on the instance $\mathbf{p}^{(j)}$. Let $N^{(j)} = \sum_{i \in [n]} N_i^{(j)}$. Note that all these numbers can be random variables where the randomness comes from coins tossed in $\mathscr{B}$.

**Lemma 19.** *Assume $\varepsilon < 0.125$ and $n$ is sufficiently large. If for every $j \in [n]$, the algorithm $\mathscr{B}$ can correctly identify the best arm in $\mathbf{p}^{(j)}$ with probability at least $0.999$ and outputs any arm for $\mathbf{p}^{(0)}$, then for some $j \in \{0, 1, \ldots, n\}$, $\mathbf{E}\left[N^{(j)}\right] \geq \frac{Cn}{\varepsilon^2}$, where $C > 0$ is a universal constant.*

Our proof of Lemma 19 is based on a reduction from a similar problem studied in [25], in which the following instances of BESTARMID have been considered:

- $\mathbf{q}^{(0)} = (q_0^{(0)}, \ldots, q_n^{(0)}) \in \mathbb{R}^{n+1}$ where for every $i \in \{0, 1, \ldots, n\}$, $q_i^{(0)} = \begin{cases} \frac{1}{2} - \varepsilon, & i = 0; \\ \frac{1}{2}, & i \neq 0. \end{cases}$

- $\forall j \in [n] : \mathbf{q}^{(j)} = (q_0^{(j)}, \ldots, q_n^{(j)}) \in \mathbb{R}^{n+1}$ where for every $i \in \{0, 1, \ldots, n\}$, $q_i^{(j)} = \begin{cases} \frac{1}{2} - \frac{\varepsilon}{2}, & i = 0; \\ \frac{1}{2} - \varepsilon, & i = j; \\ \frac{1}{2}, & \text{otherwise.} \end{cases}$

Let us fix a strategy $\mathscr{B}'$ for this collection of instances. Similarly define quantities $N_i^{(j)\prime}$ and $N^{(j)\prime}$ for $i, j = 0, 1 \ldots, n$ as we did for $\mathscr{B}$ above. The proof of [25, Theorem 1] implicitly established the following:

**Lemma 20.** *Assume $\varepsilon < 0.125$. If for every $j = 0, 1 \ldots, n$, the algorithm $\mathscr{B}'$ can correctly identify the best arm in $\mathbf{q}^{(j)}$ with probability at least $0.996$, then*

$$\mathbf{E}\left[N^{(0)\prime}\right] \geq \frac{C'n}{\varepsilon^2},$$

*where $C'$ is a universal constant.*

Armed with Lemma 20, we now prove Lemma 19.

*Proof of Lemma 19.* Assuming for the sake of contradiction that Lemma 19 does not hold, we now describe an algorithm $\mathscr{B}'$ who can correctly identify the best arm in $\mathbf{q}^{(j)}$ with probability at least $0.999$ for every $j \in \{0, \ldots, n\}$ and $\mathbf{E}\left[N^{(j)\prime}\right] < \frac{C'n}{\varepsilon^2}$ for sufficiently large $n$.

Since Lemma 19 is false, we have a promised good algorithm $\mathscr{B}$ with $C = \frac{C'}{10}$. Given any instance $\mathbf{q}^{(j)}$ with $j \in \{0, \ldots, 1\}$, we first use $\mathscr{B}$ to identify the best arm $i^*$ among arms in $\{1, 2, \ldots, n\}$. This step succeeds with probability $0.999$. Then we are left to compare arm $i^*$ with arm $0$. We pull each of the two for $K$ times and output the one with minimum practical mean. By Chernoff bound, this approach can successfully identify the best of the two with probability $0.999$ when $K = \frac{C''}{\varepsilon^2}$ for some universal constant $C'' > 0$. Therefore we have $\mathbf{E}\left[N^{(j)\prime}\right] < \frac{Cn}{\varepsilon^2} + \frac{C''}{\varepsilon^2} \leq \frac{C'n}{\varepsilon^2}$ for sufficiently large $n$ and we can identify the best arm with probability at least $0.998 > 0.996$ by the union bound. $\blacksquare$

Lemma 19 is quite general in the sense that it applies to any algorithm for BESTARMID. If we restrict our algorithm for BESTARMID to some special family of strategies, then a stronger lower bound can be obtained.

Consider the following algorithm $\mathscr{C}$: In every round, the player pulls each arm once. After $T$ rounds (so each arm has been pulled $T$ times), the player determines the best arm via the collected information. Note that we do not restrict how the player determines the best arm after collecting information for $T$ rounds, his/her strategy can be either deterministic or randomized. We prove a lower bound for $T$:

**Lemma 21.** *If for every $j \in [n]$, the algorithm $\mathscr{C}$ can correctly identify the best arm in $\mathbf{p}^{(j)}$ with probability at least $0.5$, then $T \geq \frac{\log(n/4)}{16\varepsilon^2}$.*

Note that if we apply Lemma 19 to $\mathscr{C}$, we can only get $T = \Omega\left(1/\varepsilon^2\right)$. The reason that we can obtain a stronger lower bound is the non-adaptive nature of $\mathscr{C}$.

*Proof of Lemma 21.* As a randomized algorithm can be viewed as a distribution of deterministic ones, we only need to prove the lower bound for deterministic algorithms. We prove the contrapositive of the lemma for a deterministic $\mathscr{C}$. Assume $T < \frac{\log(n/4)}{16\varepsilon^2}$. We let $W = (w_{ij})_{\substack{i \in [n] \\ j \in [T]}} \in [0,1]^{n \times T}$ be a random matrix where $w_{ij}$ is the loss of the $i$-th arm during the $j$-th pull. Our task is to study the following stochastic process, which is called *hypothesis testing* in statistics:

- Pick $J \in [n]$ uniformly at random.

- Use $\mathbf{p}^{(J)}$ to generate the matrix $W$.

- Apply $\mathscr{C}$ on $W$ to obtain $\hat{J} = \mathscr{C}(W)$.

It then follows from Fano's inequality (Lemma 17) that

$$\mathbf{Pr}\left[\hat{J} \neq J\right] \geq \frac{H(J) - I(J;W) - \log 2}{\log n} = 1 - \frac{I(J;W) + \log 2}{\log n}. \tag{4}$$

It remains to upper bound $I(J;W)$. To ease the presentation, we write $W = \left[w^{(1)}, w^{(2)}, \ldots, w^{(T)}\right]$ where each $w^{(j)} = (w_1^{(j)}, w_2^{(j)}, \ldots, w_n^{(j)})^\mathsf{T}$ for $j \in [T]$ is an $n$-dimensional column vector. It is clear that these column vectors are mutually independent *conditional on $J$*. Moreover, for each $j \in [T]$, entries in $w^{(j)}$ are mutually independent conditional on $J$ as well. Therefore, it follows from Lemma 18 that

$$I(J;W) \leq \sum_{j \in [T]} I(J;w^{(j)}) \leq \sum_{j \in [T]} \sum_{i \in [n]} I(J;w_i^{(j)}) = nT \cdot I(J;w_1^{(1)}) \leq 8\varepsilon^2 T, \tag{5}$$

where the last inequality is from a direct calculation of $I(J;w_1^{(1)})$.

Combining (4) and (5), we obtain

$$\mathbf{Pr}\left[\hat{J} \neq J\right] \geq 1 - \frac{8\varepsilon^2 T + \log 2}{\log n} > \frac{1}{2},$$

which is a contradiction. ∎

### D.3 From BESTARMID to regret

Let $G = (V, E)$ be a directed graph containing a $k$-packing independent set $S$ with $|S| \geq 2$. We assume without loss of generality that $S = \{1, 2, \ldots, |S|\}$. We construct a collection of *stochastic bandit* instances $\left\{I^{(j)}\right\}_{j \in [|S|]}$ with feedback graph $G$ as follows: For every $t \in [T]$ and $j \in [|S|]$, we use $\ell_t^{(j)}$ to denote the loss function of $I^{(j)}$ at round $t$. Let $\varepsilon \in (0,1)$ be a parameter. Then

- For all $i \notin S$, $\ell_t^{(j)}(i) = 1$;

- For $i = j$, $\ell_t^{(j)}(i) \sim \mathrm{Ber}\left(\frac{1}{2} - \varepsilon\right)$;

- For all $i \in S \setminus \{j\}$, $\ell_t^{(j)}(i) \sim \mathrm{Ber}\left(\frac{1}{2}\right)$.

Given an algorithm $\mathscr{A}$, a time horizon $T$ and $j \in [n]$, we use $R(\mathscr{A}, j, T)$ to denote the expected regret after $T$ rounds when applying $\mathscr{A}$ on $I^{(j)}$.

We show that for any $j \in [|S|]$, if an algorithm $\mathscr{A}$ has low expected regret on $I^{(j)}$, then one can turn it into another algorithm $\widehat{\mathscr{A}}$ who can identify the best arm $j$ among $S$ without exploring $S$ much.

**Lemma 22.** *Let $T \in \mathbb{N}$ be the time horizon, $\delta \in (0, 1)$ be a parameter. Let $\overline{C} = \min\{C, 1\}$ where $C$ is the constant in Lemma 19. Assuming there exists an algorithm $\mathscr{A}$ such that $R(\mathscr{A}, j, T) \leq \frac{\overline{C}\varepsilon\delta T}{4}$ for every $j \in [|S|]$, then there exists an algorithm $\widehat{\mathscr{A}}$ satisfying for every $j \in [|S|]$, if we apply $\widehat{\mathscr{A}}$ on $I^{(j)}$ for $T$ rounds, then*

- *$\widehat{\mathscr{A}}$ output $j$ with probability at least $1 - \delta$;*

- *arms in $V \setminus S$ are pulled at most $\frac{\overline{C}\varepsilon\delta T}{2}$ times in total.*

*Proof.* The algorithm $\widehat{\mathscr{A}}$ simply simulates $\mathscr{A}$ for $T$ rounds and outputs the mostly-pulled arm, breaking ties arbitrarily. We verify the two properties of $\widehat{\mathscr{A}}$ respectively.

- If the mostly-pulled arm is not $j$, then suboptimal arms must be pulled at least $\frac{T}{2}$ times. These pulls contribute at least $\frac{\varepsilon\delta T}{2} > \frac{\overline{C}\varepsilon\delta T}{4}$ expected regret.

- If arms in $V \setminus S$ are pulled more than $\frac{\overline{C}\varepsilon\delta T}{2}$ times in total, then these pulls already contribute more than $\frac{\overline{C}\varepsilon\delta T}{4}$ regret. ∎

Using the reduction in Lemma 22, we can prove Theorem 2 via lower bounds for BESTARMID.

*Proof of Theorem 2.* Assume both $|S|$ and $T$ are sufficiently large.

We first establish the $\Omega\left(\left(\frac{|S|}{k}\right)^{\frac{1}{3}} \cdot T^{\frac{2}{3}}\right)$ lower bound. Choose $\varepsilon = \left(\frac{|S|}{kT}\right)^{\frac{1}{3}}$ and $\delta = 0.001$. Suppose there exists an algorithm $\mathscr{A}$ such that $R(\mathscr{A}, j, T) \leq \frac{\overline{C}}{4000}\left(\frac{|S|}{k}\right)^{\frac{1}{3}} \cdot T^{\frac{2}{3}}$ for every $j \in [|S|]$. Then by Lemma 22, we can find an algorithm $\widehat{\mathscr{A}}$ that can correctly identify the best arm with probability at least 0.999, and observe arms in $S$ for at most $\frac{\overline{C}}{2000}|S|^{\frac{1}{3}}k^{\frac{2}{3}} \cdot T^{\frac{2}{3}}$ times (Since each pull of $V \setminus S$ observes at most $k$ arms in $S$). This contradicts Lemma 19.

Then we establish the $\Omega\left((\log|S|)^{\frac{1}{3}} \cdot T^{\frac{2}{3}}\right)$ lower bound, which needs more effort. Choose $\varepsilon = \left(\frac{\log|S|}{T}\right)^{\frac{1}{3}}$ and $\delta = 0.001$. Similar to above, suppose there exists an algorithm $\mathscr{A}$ such that $R(\mathscr{A}, j, T) \leq \frac{\overline{C}}{4000}(\log|S|)^{\frac{1}{3}} \cdot T^{\frac{2}{3}}$ for every $j \in [|S|]$. Then by Lemma 22, we can find an algorithm $\widehat{\mathscr{A}}$ that can correctly identify the best arm with probability at least 0.999, and pull arms in $V \setminus S$ for at most $\frac{\overline{C}}{2000}(\log|S|)^{\frac{1}{3}} \cdot T^{\frac{2}{3}}$ times in total.

Note that each pull of an arm $v \in V \setminus S$ in $\mathscr{A}$ observes arms in a subset $S' \subseteq S$. The key observation is that we can assume without the loss of generality that $S' = S$, since this assumption only increases the power of the algorithm. This assumption can make our algorithm non-adaptive.

We rigorously prove this using a coupling argument. Assume we design an algorithm $\widehat{\mathscr{B}}$ in which each pull of an arm in $V \setminus S$ can observe all arms in $S$. We show that $\widehat{\mathscr{B}}$ can perfectly simulate $\widehat{\mathscr{A}}$, and therefore $\widehat{\mathscr{B}}$ is more powerful. So if we have a lower bound for $\widehat{\mathscr{B}}$, it is automatically a lower bound for $\widehat{\mathscr{A}}$. Note that the behavior of an algorithm for BESTARMID in each round is determined by the information collected and coins tossed so far. If we apply both $\widehat{\mathscr{A}}$ and $\widehat{\mathscr{B}}$ to some $I^{(j)}$ and assume (1) both algorithms use the same source of randomness; and (2) all the loss vectors $\ell_t^{(j)}$ use the same source of randomness, then the information collected by $\widehat{\mathscr{A}}$ at any time is a subset of that of $\widehat{\mathscr{B}}$. Therefore we can use $\widehat{\mathscr{B}}$ to simulate $\widehat{\mathscr{A}}$ and outputs the same as $\widehat{\mathscr{A}}$.

Based on this observation, we use the non-adaptive nature of $\widehat{\mathscr{B}}$ and apply Lemma 21 to get an immediate contradiction. ∎

# E   Time-varying graphs

In this section, we will consider time-varying graphs. Instead of a fixed graph $G = (V, E)$ for all $T$ rounds, for each round $t \in [T]$, we obtain an observable graph $G_t = (V, E_t)$. We slightly abuse notations here and let $G = (G_t)_{t \in [T]}$ and $R(G, T)$ denote the min-max regret for time-varying graphs $G$.

**Corollary 23.** *The min-max regret for sequential time-varying graphs $G = (G_t)_{t \in [T]}$ satisfies*

$$R(G, T) = O\left( \left( \overline{\delta^*}(G) \log n \right)^{\frac{1}{3}} T^{\frac{2}{3}} \right),$$

*where $\overline{\delta^*} \triangleq \frac{\sum_{t=1}^{T} \delta_t^*(G_t)}{T}$ is the average fractional weak domination number for graphs $G$ in $T$ rounds.*

*Proof.* We can slightly modify Algorithm 1: In each round $t$, we use $G_t$ as an input instance to calculate $\hat{\ell}_t$ and $\mathbf{u}_t$. Then the left proof is totally similar to the proof of Theorem 1 except for replacing (3) with $Regret \leq \frac{\log n}{\eta} + \eta n T + \frac{\eta \sum_{t=1}^{T} \delta^*(G_t)}{2\gamma} + \gamma T = \frac{\log n}{\eta} + \eta n T + \frac{\eta \bar{\delta}^* T}{2\gamma} + \gamma T.$ ∎

A similar case is the probabilistic graph model. A probabilistic graph can be denoted as a triple $\mathcal{G} = (V, E, P)$ where $P : E \to (0, 1]$ assigns a *triggering probability* for each edge in $E$. In each round $t$, a realization of $\mathcal{G}$ is a graph $G_t = (V, E_t)$ where $E_t = \{e \in E : O_{t,e} = 1\}$ and here $O_{t,e}$ is an independent Bernoulli random variable with mean $P(e)$. Note that here we need to ensure that any realization of $\mathcal{G}$ is an observable graph. By abuse of notations, we denote by $G = (G_t)_{t \in [T]}$ the sequential realizations of $\mathcal{G}$. Define $R(\mathcal{G}, T) = \inf_{\mathscr{A}} \sup_{\ell^*} \mathbf{E}\left[ \sum_{t=1}^{T} \ell_t(A_t) - \ell_t(a^*) \right]$ as the min-max regret for the probabilistic graph $\mathcal{G}$ and here the randomness comes from the algorithm and sequential graphs $G$.

**Corollary 24.** *The min-max regret for the probabilistic graph $\mathcal{G}$ satisfies*

$$R(\mathcal{G}, T) = O\left( \left( \mathbf{E}\left[ \delta^*(G_1) \right] \log n \right)^{\frac{1}{3}} T^{\frac{2}{3}} \right),$$

*Proof.*

$$R(\mathcal{G}, T) = \mathbf{E}\left[ \sum_{t=1}^{T} \ell_t(A_t) - \ell_t(a^*) \right] = \mathbf{E}\left[ \mathbf{E}\left[ \sum_{t=1}^{T} \ell_t(A_t) - \ell_t(a^*) \,|\, G \right] \right]$$

$$\overset{(\diamondsuit)}{=} O\left( \mathbf{E}\left[ \left( \frac{\sum_{t=1}^{T} \delta^*(G_t)}{T} \right)^{\frac{1}{3}} \right] (\log n)^{\frac{1}{3}} T^{\frac{2}{3}} \right) \overset{(\clubsuit)}{=} O\left( \left( \mathbf{E}\left[ \delta^*(G_1) \right] \log n \right)^{\frac{1}{3}} T^{\frac{2}{3}} \right),$$

where $(\diamondsuit)$ follows from Corollary 23 and $(\clubsuit)$ comes from the Jensen inequality and the independence of $(G_t)_{t \in [T]}$. ∎

# F   Numerical experiments

According to Theorem 1, our algorithm 1 will outperform Exp3.G in [1] when $\delta^* \ll \delta$. Therefore, we run our experiments on graphs $G = (V_1 \cup V_2, E)$ with orthogonal relation on $\mathbb{F}_2^k$.

We choose $k = 2, 3, 4, 5$ and set $T = 20 \times n^3 \log(n)/\delta^{*2}(G)$, $\gamma = \left( \frac{\delta^*(G) \log n}{T} \right)^{1/3}$, and $\eta = \frac{\gamma^2}{\delta^*(G)}$ for our algorithm. The similar parameters are set for Exp3.G by replacing $\delta^*$ with $\delta$. Our adversary is *nonoblivious* which means the loss vector $\ell_t$ is allowed to depend on $\mathcal{F}_{t-1}$. Concretely, for each time $t$, the adversary will see the distribution $X_t$ and find the vertex $i$ in $V_2$ with the minimum $X_t(i)$. Then the adversary provides $i$ with loss 0 and loss 1 for all other vertices.

We can immediately see from Figure 2 that our algorithm always outperforms Exp3.G when $t$ is large compared to $T$.

According to Appendix C.3, $\delta^* = \frac{2^k - 1}{2^{k-1}}$ and $\delta = k$. Table 2 shows the experimental ratio of our algorithm's regret to Exp3.G's in the terminating time $T$. The ratio is positively linearly related to $k$, which supports our theory result $\frac{\delta}{\delta^*}$ to some extent.

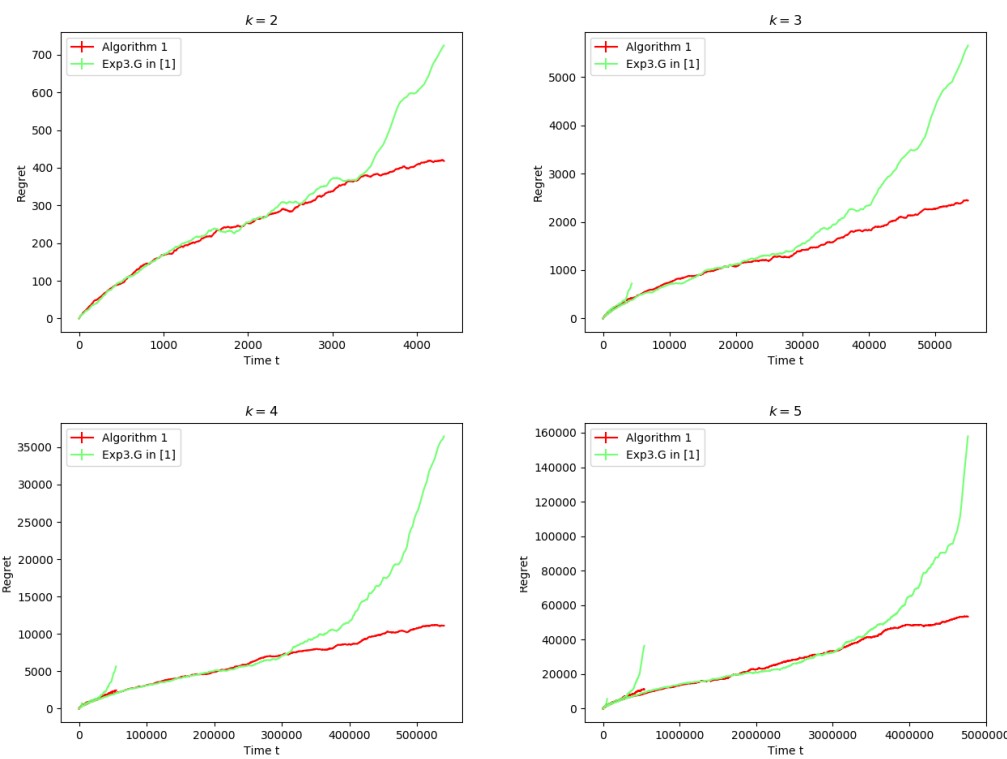

Figure 2: Regret comparison for different sizes of graphs

Table 2: Regret ratio of two algorithms

| $k$ | 2 | 3 | 4 | 5 |
|---|---|---|---|---|
| regret of Exp3.G | 724.6 | 5656.3 | 36461.5 | 157909.5 |
| regret of Algorithm 1 | 417.9 | 2443.0 | 11115.1 | 53305.8 |
| ratio | 1.73 | 2.32 | 3.28 | 2.96 |