# OpenReview forum: "Understanding Bandits with Graph Feedback"
_NeurIPS.cc/2021/Conference — NeurIPS 2021 Poster_

### Official Review · Reviewer_DW8T · 2021-07-15

**Rating:** 6
**Confidence:** 3

**Summary:**

This paper connects notions of the fractional weak domination number and the k-packing independence number with the graph bandits problem. Under these notions, the authors propose a new algorithm that achieves a fractional weak domination number dependent upper bound and also constructs lower bounds, including the main meta-lower bounds and other lower bounds in terms of different classical graph parameters. Therefore, tighten the gap between the upper and lower bound compared to the previous works.

**Limitations And Societal Impact:**

The authors have adequately addressed the limitations

**Main Review:**

--Originality:
For the lower bound analysis, the technique to construct lower bound for regret via best-arm-identification problems is standard. One innovation here is that the authors use k-packing independence numbers to adopt these lower bound construction techniques into a graph structure.  Then by showing the relationship between the 1-packing independent set and the solution of the integral program defined in the paper, the author gets a fractional weak domination number dependent bound (Thm 3). The technique used in Thm3 seems purely related to graph theories. It looks interesting to me but I am not sure how novel it is in the graph theory area.

For the upper bound analysis, the choice of OMD framework, regularization functions, the estimator, and the exploration parameters are all standard techniques. One small innovation is that the exploration parameter u comes from the solution of a linear program, instead of simply a uniform distribution over some representative set.

Overall, I think this paper combines some well-known techniques from online learning and graph theory (or linear programming). The combination techniques themselves have some novelty.

--Quality:
All claims are well supported

--Clarity:
This paper is well written

--Significance:
This paper closes the upper bound and lower bound gaps for graph bandits and also discusses their results in several classical graph structures. So I think from a theoretical perspective, this contribution is meaningful.

The technique's novelty here is nontrivial but somehow limited.

--Questions: I wonder if this algorithm can directly extend to more adaptive settings [21, 5,19...] by combining with those existing techniques listed in the related work? If so that would be attractive.


**Time Spent Reviewing:**

5

---

> ### Author Response · Authors · 2021-08-09
> **Regarding concerns in the comments**
>
> Dear reviewer, thanks for the comments. Yes, our proof of Thm 3 is the main technical contribution of the work. The graph theory part argument is tailored for the proof here and to the best of our knowledge, the method is new to the community.
>
> Regarding your question on the extension to adaptive settings, well, as far as we can see, a direct extension using our new graph parameter seems to be difficult and some new idea is required. This is a very interesting question for further investigation.

---

### Official Review · Reviewer_9yjH · 2021-07-16

**Rating:** 7
**Confidence:** 4

**Summary:**

This paper studies the min-max regret bounds of the adversarial bandit with graph feedback. The graph feedback means that when pulling an arm, the learner can also get feedback from the arms that are adjacent to this arm in a specific graph $G$. The authors made non-trivial improvements on the upper bound and lower bound on the regret, especially for the “weakly observable” graphs. For some special cases such as complete
bipartite graphs and orthogonal relation on $\mathbb{F}^k_2$, the regret upper and lower bounds are optimal up to constant factors.


**Limitations And Societal Impact:**

Limitation: The improvements made by the authors do not cover most of the general cases, which has been discussed in the box Main Review.

I do not find negative societal impact of this work as this work is pure-theory and focuses on a fundamental abstract problem.

**Main Review:**

In summary, the problem is significant for the subarea of graphical bandits, and the authors made non-trivial contributions to this problem, though there are still rooms for further improvements.

Originality: The techniques and methods used in this paper are novel. By making a deeper analysis on the problem and introducing the parameter $\delta^*(\cdot)$ of a graph $G$, the authors were able to show a better bound compared to $\delta(\cdot)$ that are commonly adopted in previous works (note $\delta^*(G) \leq \delta(G)$ for any graph $G$).

Quality: Overall, this paper made non-trivial contributions to this problem and the introduction of $\delta^*(\cdot)$ can inspire future studies. In my opinion, it may be hard to make further improvements without using new techniques. Thus, although in general, there is still a $\frac{\log{n}}{\log\log{n}}$ gap, I think the results in this paper are good enough, especially for some special cases where the upper bound matches the lower bound (ignoring constant factors).

Clarity: The paper is well written overall but has some typos and formatting issues. For instance, Table 1 is surpassing the margin and needs to be restructured, and the reference in Line 63 is ??.

Significance: The problem is significant, but this paper only makes significant contributions to some special cases. For General cases, the improvements are limited.

In conclusion, my score lies between 6 and 7. I tend to give a score 6 for now due to the lack of practical motivations and numerical results. A typical motivation can be utilizing the information or data of the queried user's friends or connections, but this application may have potential privacy issues. It may help the readers if the authors can add more practical motivations and applications.

-----------------------------------------------------------------
After reading the response, the score is increased from 6 to 7.

**Time Spent Reviewing:**

2

---

> ### Author Response · Authors · 2021-08-09
> **Practical motivations and the significance of the results**
>
> Dear reviewer, thanks for the comments. We will say some words about the practical motivation of this work and discuss why our theoretical results are useful.
>
> Bandits with graph feedback usually captures online-learning scenarios with side information available. Some notable applications include web advertising (see [24, Mannor and Shamir, NeurIPS 2011] for details), video recommendation, etc.
>
> In web advertising, the vertex set of the feedback graph is all ads and the edges characterize the correlation between ads. Suppose ad $i$ stands for running shoes, ad $j$ stands for a wheelchair. Then a user who clicks ad $i$ is unlikely interested in $j$ and vice versa. So in the feedback graph, we have an undirected edge between $i$ and $j$.
>
> Another potential application is video recommendation in some popular short-video apps (for example, TikTok). Every time the app starts, it plays a video that hopes to capture your attention. One way to improve the quality of the start-up recommendation is to show some similar videos alongside and ask the user to pick some to watch next. The recommendation algorithm can then learn whether those similar videos are liked by the user. Therefore, we let the vertex set of the feedback graph be all videos and add an edge between two videos if they are similar. This is a weakly observable graph.
>
> In fact, in both applications described above, the number of edges incident to a specific vertex (either correlated ads or similar videos) is small. Therefore our assumption in Corollary 4, the graph is of bounded in-degree, is reasonable. In this setting, our algorithm improves previous results and is optimal up to an $O((\log n)^{\frac{1}{3}})$ factor.
>
> Although our paper is theoretically oriented and mainly aims at narrowing gaps between upper and lower bounds, we will experiment with our algorithm in the revised version of the paper. Thanks for your suggestion.

---

> > ### Comment · Reviewer_9yjH · 2021-08-20
> > **Increase score to 7 after reading the response**
> >
> > Thanks for the response. The practical motivations look reasonable and the authors promise to add experiments, so I am increasing the score to 7.

---

### Official Review · Reviewer_3V7a · 2021-07-17

**Rating:** 6
**Confidence:** 4

**Summary:**

The authors consider bandits with graph feedback and show improved lower and upper regret bounds involving a new quantity: fractional weak domination number. This leads to some logarithmic dependence improvement on the number of nodes.

**Main Review:**

- The paper is well written and well organized, which provides sufficient background knowledge to reach the results. The table is also helpful.
- For the upper bounds, I think it is interesting to see the improvement, but overall the the algorithm is not very novel, which seems to be still Exp3.G but with the fractional exploration probability. Related to this, I think the authors should compare their algorithm with Exp3.G or other bandit with graph feedback algorithms. The unbiased estimator and the negative entropy function are both standard in this field I think. The only difference as I mentioned should be the construction of u. The current presentation may give readers an impression that the algorithm is all new in these parts.
- Moreover, the improvement from \delta to \delta* seems a little bit incremental to me. One thing I think is more interesting but the authors didn't even mention is about the computational efficiency. I think in the original Exp.G algorithm, we need to have a weak domination set to achieve such bounds, but the set is NP-hard to get in general. However, the exploration probability the authors propose can be efficiently computed via a linear program. Maybe the authors can verify whether this is the case, and it is nice to discuss this in the future revision.
- As for the lower bounds, although the authors say "we can not find an instance such that our lower bound is worse than the previous one", this claim looks a little bit weak to me and I may still think the bounds are in general incomparable and the improvement in some special instances seems not very significant.
- What is BESTARMID around Line 351? I couldn't find it in the main text.
- Around Line 118 the authors mention [35] and say [35] gives another way to achieve the optimal regret bound. But what theorem or algorithm are the authors referring to? I thought the paper is mainly about adaptive bounds and many results in the paper have logarithmic terms in T, which may not be considered "optimal" here.

Overall, I think the authors have an interesting observation on this problem, but the results are not a very significant improvement compared to the previous works. Also, there is not much algorithmic contribution. Therefore, I give the marginally below the acceptance threshold rating to this paper.

===========

I thank the authors for the valuable discussions and I increase my score based on this.

**Time Spent Reviewing:**

5

---

> ### Author Response · Authors · 2021-08-09
> **A detailed reply to the concerns**
>
> Dear reviewer, thanks for your detailed comments. Please find our replies to your concerns below.
>
>
> **Why not compare the algorithm with Exp3.G?**
>
> Yes, our algorithm is equivalent to Exp3.G in [Alon et al. 2015] with a different exploration probability in terms of a new graph parameter $\delta^\ast$. We mentioned in the paper that our main novelty in the upper bound is this new graph parameter (line 48-50), and the whole algorithm is the standard OSMD with negative entropy as the potential function (line 201-201). Nevertheless, we will compare and discuss the connection and difference between our algorithm and Exp3.G in the revised version. Thanks.
>
>
> **The improvement from $\delta$ to $\delta^\ast$ seems a little bit incremental...**
>
> Well, although it is relatively easy to adapt both the algorithm and the analysis of [Alon et al. 2015] to obtain upper bound in terms of $\delta^\ast$, we believe that the observation here is important and non-trivial for the following two reasons.
>
> First, as mentioned in the introduction, the ultimate goal of this line of research is to identify the *correct* graph parameter to characterize the mini-max regret. We believe and give evidences to show that $\delta^\ast$ is a reasonable candidate as matching lower bound has been proved for some instances. Therefore, the step from $\delta$ to $\delta^\ast$ is crucial and might be already tight.
>
> Secondly, as $\delta^\ast$ is the optimal solution of a certain LP, this allows us to prove lower bounds in a more natural way: applying the strong duality theorem and using a new rounding method to construct hard instances from fractional solutions of the dual LP. This observation explains why a lower bound in terms of $\delta^\ast$ should exist and it is also the main technical contribution of this work. Some improved and optimal lower bounds are obtained in this way and we think this method towards lower bound is new to the community.
>
>
> **...the exploration probability the authors propose can be efficiently computed via a linear program. Maybe the authors can verify whether this is the case...**
>
> Yes, this is the case and thanks for pointing it out. Our exploration probability can be efficiently computed by solving the corresponding LP while it is NP-hard to determine the exploration probability used in [Alon et al. 2015]. We will mention this in the revised version.
>
>
> **...I may still think the bounds are in general incomparable and the improvement in some special instances seems not very significant...**
>
> Some of the special instances we studied in the paper are important both in theory and in practice. For example, we narrow the gap for graphs of bounded in-degree graphs in Corollary 4. In fact, this is the case for some notable applications of bandits with graph feedback including web advertising, video recommendation, etc.
>
> In web advertising (see [24, Mannor and Shamir, NeurIPS 2011] for details), the vertex set of the feedback graph is all ads and the edges characterize the correlation between ads. Suppose ad $i$ stands for running shoes, ad $j$ stands for a wheelchair. Then a user who clicks ad $i$ is unlikely interested in $j$ and vice versa. So in the feedback graph, we have an undirected edge between $i$ and $j$.
>
> Another potential application is video recommendation in some popular short-video apps (for example, TikTok). Every time the app starts, it plays a video that hopes to capture your attention. One way to improve the quality of the start-up recommendation is to show some similar videos alongside and ask the user to pick some to watch next. The recommendation algorithm can then learn whether those similar videos are liked by the user. Therefore, we let the vertex set of the feedback graph be all videos and add an edge between two videos if they are similar. This is a weakly observable graph.
>
> In both applications described above, the number of edges incident to a specific vertex (either correlated ads or similar videos) is small. Therefore our assumption in Corollary 4, the graph is of bounded in-degree, is reasonable.
>
>
> **What is BESTARMID around line 351? \& The [35] cannot achieve the optimal regret bound.**
>
> Sorry that we forgot to define the BESTARMID problem in the main text. It is first mentioned in Section D.2 in the full version of the paper. We also confirm that the [35] can only get an upper bound $\sqrt{nT\log T}$. We will fix these two problems in the revised version.

---

> > ### Comment · Reviewer_3V7a · 2021-08-13
> > **Thanks for the responses**
> >
> > I thank the reviewers for their detailed responses. I agree that even though the improvement from $\delta$ to $\delta^*$ might be incremental in terms of the bounds, it is probably more meaningful for the community to know the right graph parameter. For the lower bound, I think my main concern was the lower bound is not strictly better than the previous one. However, on the second thought, I think this is probably not a big issue for lower bounds because you can always pick the better one and get the best of the both bounds (unlike upper bounds, for which this process may not be trivial). I think the authors may can confirm whether what I say is correct, and it is nice to clarify this in the future revision.
> >
> > For the video recommendation example, I am not sure I understand the setting. Specifically, I don't get why we don't have self-loop here. It would be great if the authors can elaborate on this.

---

> > > ### Author Response · Authors · 2021-08-13
> > > **clarification, and more on video recommendation**
> > >
> > > Thanks for the reply! Yes, for a specific instance, one can always pick the best lower bounds, no matter how it is proved. We will clarify this in the revised version. Thanks!
> > >
> > > About the video recommendation, since the video played at the start of the app is recommended by the algorithm, but not triggered by the user's click, we believe that it is less informative for the user to specify whether he/she likes it. In other words, when a random video is presented to a user with a **Like Button**, the behavior of the user is somehow arbitrary and he/she may *not* click that button with some probability even if he/she likes it. So we don't add self-loop to the recommended video. On the other hand, we believe that whether the user picks and watches a video in a given list can reflect his/her preference more accurately.
> > >
> > > Nevertheless, *weakly observable feedback graphs* allows self-loops, as long as not every node has one. So it provides a more flexible way to model practical applications.

---

### Official Review · Reviewer_LkFC · 2021-07-18

**Rating:** 7
**Confidence:** 4

**Summary:**

This paper studies regret-bounds for semi-bandits in the feedback graph model for weakly observable feedback graphs, and (sometimes) improves both the upper and lower bounds on the regret in the adversarial setting (though in general the bounds are incomparable). In particular, they characterize the regret more precisely than existing work by going beyond weak domination number regret bounds in this setting; they use the notion of the fractional weak domination number (previously used for the case of strongly observable feedback graphs) and the k-packing independence number and show bounds in terms of these parameters. Since the gap between the upper and lower bounds on the regret depends on the integrality gap for a linear program related to the feedback graph (corresponding to the fractional weakly dominating sets problem on the graph), they then provide some natural concrete instances (e.g. bounded degree graphs) where the lower bound is now tight up to a factor of $\log^{1/3}(n)$.

**Ethical Concerns:**

No ethical concerns.

**Limitations And Societal Impact:**

Yes, as this is a theory paper.

**Main Review:**

This work follows an interesting line of work in proving regret bounds for the adversarial semi-bandit problem in terms of parameters of the feedback graph. The results are novel and interesting, and are definitely worth of acceptance. The paper is well-written and clearly contextualizes their results compared to the prior work. Moreover, some new techniques are introduced to get the lower bound -- the proof initially follows the same outline as in the original semi-bandit feedback paper lower bound, but is able to conduct a smarter analysis by reducing to the problem of best arm identification (e.g. Thm.2, which I found interesting and useful).  They also give a novel rounding procedure on the aforementioned LP to connect the integrality gap to k-packing independence number ( which shows up in Thm 2), thus completing the connection to the upper bound results. The upper bound result is almost standard; the only novelty here is the choice of exploration distribution, which depends on rounding the LP (the analysis is standard and the dependence on the integrality gap falls out by definition).

Overall, I found the results of section D.2 and D.3 the most interesting and useful technical part of the paper, in addition to the actual result (identification of graph parameters that better characterize the regret in the weakly observable setting, and even more tightly in a few natural concrete instances).  I would encourage some re-writing of lines 741-750 -- the logical flow of the argument is a bit disjointed, and could read more smoothly given that the argument is very simple. Overall, I recommend acceptance.

**Time Spent Reviewing:**

3 hours

---

> ### Author Response · Authors · 2021-08-09
> **Re-write some proof to improve presentation.**
>
> Dear reviewer, thanks for the comment! We will re-write line 741-750 to improve the presentation in the revised version.

---

### Decision · Program_Chairs · 2021-09-28

**Decision:**

Accept (Poster)

**Comment:**

Several reviewers raised their scores after discussing with the authors.
Overall, we agree that the contribution of this paper is significant enough for better understanding the
minimax regret of bandits with graph feedback.
Please do incorporate all the suggestions from the reviewers into the final version.

**Consistency Experiment:**

NeurIPS has a long history of experimentation. In 2014, NeurIPS ran an experiment in which 10% of submissions were reviewed by two independent committees to quantify the randomness in the review process. This year, we repeated a variant of this experiment to see how the quality of the review process has changed over time.  This paper was part of the experiment and was therefore assigned to two committees (consisting of reviewers, an Area Chair, and a Senior Area Chair) that reached independent decisions.  If both committees made the same recommendation, this recommendation was followed. If a single committee recommended acceptance, the paper was accepted (with the exception of a few cases in which the other committee identified what we considered a fatal flaw, e.g., an error in a key result).

Both committees reached the same decision: **Accept (Poster)**

The other committee assigned to the paper recommended **Accept (Poster)**.  You can find the other set of reviews, along with any follow up discussion with the authors here:
https://openreview.net/forum?id=V3aZTKsHykQ